

# Reconstruction of multi-millennial summer climate variations in central Japan by integrating tree-ring cellulose oxygen and hydrogen isotope ratios

Takeshi Nakatsuka[1,2], Masaki Sano[1,3], Zhen Li[1,2], Chenxi Xu[1,4], Akane Tsushima[1], Yuki Shigeoka[2],
Kenjiro Sho[5], Keiko Ohnishi[6], Minoru Sakamoto[7], Hiromasa Ozaki[7,8], Noboru Higami[9], Nanae Nakao[10,11],
Misao Yokoyama[12] & Takumi Mitsutani[13]

[1] Research Institute for Humanity and Nature, Kyoto 603-8047, Japan.
[2] Graduate School of Environmental Studies, Nagoya University, Nagoya 464-8601, Japan.
[3] Faculty of Human Sciences, Waseda University, Tokorozawa 359-1192, Japan.
[4] Key Laboratory of Cenozoic Geology and Environment, Institute of Geology and Geophysics, Chinese Academy of Sciences, Beijing 100029, China.
[5] Department of Architecture, Civil Engineering and Industrial Management Engineering, Nagoya Institute of Technology, Nagoya 466-8555, Japan.
[6] Institute of Low Temperature Science, Hokkaido University, Sapporo 060-0819, Japan.
[7] National Museum of Japanese History, Sakura 285-8502, Japan.
[8] The University Museum, The University of Tokyo, Tokyo 113-0033, Japan.
[9] Aichi Prefectural Center for Archaeological Operations, Yatomi 498-0017, Japan.
[10] Research Center, Musashi University, Tokyo 176-8534, Japan.
[11] Faculty of Science, Yamagata University, Yamagata 990-8560, Japan.
[12] Graduate School of Agriculture, Kyoto University, Kyoto 606-8502, Japan.
[13] Nara National Research Institute for Cultural Properties, Nara 630-8577, Japan.

*Correspondence to*: Takeshi Nakatsuka (nakatsuka.takeshi@f.mbox.nagoya-u.ac.jp)

**Abstract.** Oxygen isotope ratios ($\delta^{18}O$) of tree-ring cellulose are a novel proxy of summer hydroclimate in monsoonal Asia. In central Japan, we collected 67 conifer wood samples, mainly *Chamaecyparis obtusa*, with ages encompassing the past 2,600 yr. The samples were taken from living old trees, excavated archeological wood, old architectural wood, and naturally buried logs. We analyzed stable isotope ratios of oxygen ($\delta^{18}O$) and hydrogen ($\delta^2H$) in tree-ring cellulose in these samples without using a pooling method, and constructed a statistically reliable tree-ring cellulose $\delta^{18}O$ time-series for the past 2,500 yr. However, there were distinct age trends and level offsets in the $\delta^{18}O$ record, and cellulose $\delta^{18}O$ values showed a gradual decrease as an individual tree matures. This suggested it is difficult to establish a cellulose $\delta^{18}O$ chronology for low-frequency signals by simple averaging of all the $\delta^{18}O$ time-series data. However, there were opposite age trends in the cellulose $\delta^2H$, and $\delta^2H$ gradually increased with tree age. There were clear positive correlations in the short periodicity variations between $\delta^{18}O$ and $\delta^2H$, probably indicating a common climate signal. A comparison of the $\delta^{18}O$ and $\delta^2H$ time-series in individual trees with tree-ring width suggested that the opposite age trends of $\delta^{18}O$ and $\delta^2H$ are caused by temporal changes in the degree of post-photosynthetic isotope exchange with xylem water, accompanied by changes in stem growth rate (growth effect) that are influenced by human activity in the forests of central Japan. Based on the assumptions that cellulose $\delta^{18}O$ and $\delta^2H$ vary





positively and negatively with constant proportional coefficients due to climate variations and the growth effect, respectively, we solved simultaneous equations for the climatological and physiological components of variations in tree-ring cellulose $\delta^{18}O$ and $\delta^2H$ in order to remove the age trend (growth effect). This enabled us to evaluate the climatic record from cellulose $\delta^{18}O$ variations. The extracted climatological component in the cellulose $\delta^{18}O$ for the past 2,600 yr in central Japan was well

correlated with numerous instrumental, historical, and paleoclimatological records of past summer climate at various spatial and temporal scales. This indicates that integration of tree-ring cellulose $\delta^{18}O$ and $\delta^2H$ data is a promising method to reconstruct past summer climate variations on annual to millennial time-scales, irrespective of the growth affect. However, analytical and statistical methods need to be improved for further development of this climate proxy.

## 45  1  Introduction

In general, it is not straightforward to extract low-frequency climate signals from tree-ring time-series, because there are typically age trends in tree-ring width. To overcome this, the regional curve standardization (RCS) method is used to determine the typical growth curve in a region for a tree species, which is subtracted from the tree-ring width time-series of individual trees to estimate climatological influences explicitly (Esper et al., 2003).

Tree-ring cellulose oxygen isotope ratios ($\delta^{18}O$) are a novel proxy of past summer hydroclimate, and it has been reported that such data are not affected by age trends (Young et al., 2011; Kilroy et al., 2016; Xu et al., 2016) or only show short-lived juvenile effects of higher $\delta^{18}O$ values (Szymczak et al., 2012; Duffy et al., 2017). These juvenile effects are interpreted to be related to the less developed root system of young trees and the evaporative enrichment of $^{18}O$ in soil water in the near-surface. However, Esper et al. (2010) demonstrated the presence of long-term age trends, whereby $\delta^{18}O$ gradually decreases for

hundreds of years after germination, as well as significant level offsets between different individual trees, in a conifer species. This requires three analytical criteria to evaluate the age trends and level offsets in the tree-ring cellulose $\delta^{18}O$ data: (1) well-replicated data; (2) non-pooled data; and (3) composite chronologies from multiple species and regions. However, given that subsequent studies of tree-ring cellulose $\delta^{18}O$ have reported that age trends are negligible, the mechanisms responsible for long-term age trends remain unclear.

In Japan, it has been difficult to establish millennial-scale tree-ring chronologies applicable to reconstructing past changes in summer climate, even though such information is critical for rice paddy cultivation and for historical and archeological studies in Japan. Although there are ancient archeological and architectural woods that have been used for dendrochronological analysis of wooden artifacts with ages of 1,000–2,000 yr (Mitsutani, 2000), these have not been utilized for past climate reconstructions. This is because tree-ring widths measured in trees from normal Japanese forests do not have a high sensitivity

to summer climate. Moreover, the ecological disturbance of tree growth by neighboring trees in dense Japanese forests prevents reconstructions of past climate based on the relatively small number of old sub-fossil wood samples that are available.



However, tree-ring cellulose $\delta^{18}O$ data are sensitive to summer hydroclimate in Asia (Treydte et al., 2006; Grießinger et al., 2011; Li et al., 2011; Xu et al., 2011; Sano et al., 2012), including in Japan (Nakatsuka et al., 2004; Tsuji et al., 2008; Yamaguchi et al., 2010; Li et al., 2015). High correlation coefficients for tree-ring cellulose $\delta^{18}O$ data amongst different individual trees and species make it possible to reconstruct past summer climate using only a small number of sub-fossil trees. Therefore, during the past decade we have analyzed tree-ring cellulose $\delta^{18}O$ in various types of conifer wood with large tree-ring numbers (mainly *Chamaecyparis obtusa*), including from living old trees, excavated archeological wood, old architectural wood, and naturally buried logs, in order to reconstruct past variations in summer hydroclimate in central Japan.

We do not use the pooling method (Szymczak et al., 2012; Loader et al., 2013) for the tree-ring isotope measurements. Instead, we analyze all the tree-rings in the wood samples using a novel method, involving direct cellulose extraction from 1-mm-thick wood sections (Kagawa et al., 2015). This allows us to evaluate age trends and level offsets in the tree-ring cellulose $\delta^{18}O$ data of individual trees, following the criteria proposed by Esper et al. (2010). This approach has shown that most trees are significantly affected by long-term age trends and level offsets, such that it is not possible to reconstruct long-term summer climate variations without taking these effects into accounts.

However, we have also found that hydrogen isotope ratios ($\delta^2H$) of tree-ring cellulose, which have been measured simultaneously with the $\delta^{18}O$ data, have age trends opposite to those of $\delta^{18}O$, and that $\delta^2H$ is positively and negatively correlated with $\delta^{18}O$ in terms of short- and long-periodicity variations, respectively. This $\delta^{18}O$ and $\delta^2H$ relationship provides insights into the mechanisms responsible for age trends in tree-ring cellulose $\delta^{18}O$ data, and makes it possible to remove the age trends in such data.

Here, we compare $\delta^{18}O$ and $\delta^2H$ data for tree-ring cellulose, and elucidate the mechanisms responsible for age trends in tree-ring cellulose $\delta^{18}O$ data. We then propose and apply a new method to the tree-ring cellulose $\delta^{18}O$ and $\delta^2H$ time-series for the past 2,600 yr in central Japan. This enables us to remove the age trend and extract the climatological component of the tree-ring cellulose $\delta^{18}O$ data on 1–1,000 yr time-scales.

## 2 Samples and chemical analyses

### 2.1 Collection of tree-ring samples

A total of 67 tree samples covering the past 2,600 yr were collected in central Japan (Fig. 1). These samples consist of living old trees, old architectural wood, excavated archeological wood, and naturally buried logs (Table 1). Most of the trees are Japanese cypress (*Chamaecyparis obtusa*), but there are some samples of other conifer species (*Chamaecyparis pisifera*, *Sciadopitys verticillata*, and *Cryptomeria japonica*). All samples contain hundreds of tree-rings. Although some samples have already been dated by traditional dendrochronological methods using tree-ring width (Mitsutani et al., 2000), the new tree-ring $\delta^{18}O$ time-series were used in the final dendrochronological determination of their ages (Roden, 2008; Yamada et al., 2018), which provided results consistent with the tree-ring width dating results.





## 2.2 Measurement of tree-ring cellulose δ$^{18}$O and δ$^2$H

Cellulose is extracted from tree-rings using the method of Kagawa et al. (2015). Firstly, 1-mm-thick cross-sectional laths
perpendicular to the cellulose fibers were sliced from all wood samples and directly subjected to chemical treatment to remove
components other than cellulose. A 100–300 μg fragment of cellulose representing the whole annual layer was then cut from
the cellulose lath year-by-year using a fine blade under a microscope. Each sample was wrapped in silver foil for δ$^{18}$O and δ$^2$H
measurements using a mass spectrometer combined with a pyrolysis elemental analyzer (TCEA/Delta V Advantage;
Thermofisher Scientific, Bremen, Germany). After measurements of every eight samples, a cellulose standard (cellulose
reagent; Merck) with known δ$^{18}$O and δ$^2$H was analyzed to calibrate analytical drift and calculate the δ$^{18}$O and δ$^2$H of samples
against the international standard of VSMOW, following Eqs (1) and (2). Analytical reproducibility (1σ) based on repeated
measurements of homogeneous cellulose samples is ca. ±0.15‰ and ±1.5‰ for δ$^{18}$O and δ$^2$H, respectively.

$$\delta^{18}O_{vsmow} = \left\{ \frac{\left(^{18}O/^{16}O\right)_{sample}}{\left(^{18}O/^{16}O\right)_{vsmow}} - 1 \right\} \times 1000(‰), \tag{1}$$

$$\delta^{2}H_{vsmow} = \left\{ \frac{\left(^{2}H/^{1}H\right)_{sample}}{\left(^{2}H/^{1}H\right)_{vsmow}} - 1 \right\} \times 1000(‰). \tag{2}$$

Given that cellulose contains 30% of OH-group hydrogen that is exchangeable with experimental water during chemical
treatment, it is necessary to remove the OH-group by nitration of the cellulose (synthesis of nitrocellulose) before the isotope
measurements (Epstein et al., 1976; DeNiro, 1981). It was not possible to nitrate all the cellulose samples analyzed in this
study. However, we can expect that all OH-group hydrogen in a cellulose lath will have a unique δ$^2$H value, as it would have
been replaced by the hydrogen of homogeneous water in the test tube during the process of cellulose extraction, such that the
measured tree-ring cellulose δ$^2$H time-series will retain the original isotopic variations, although the amplitude of these
variations will decrease by up to 70% and the absolute values will have been modified (Filot et al., 2006). We confirmed that
there are consistent correlations in δ$^2$H variations between cellulose and nitrocellulose produced using the traditional nitration
method (Epstein et al., 1976; DeNiro, 1981) for two tree samples (Fig. 2). This demonstrates that it is possible to reconstruct
original variations in tree-ring cellulose δ$^2$H data without nitrating the cellulose. However, our δ$^2$H measurements are
significantly influenced by a memory effect from the previous few measurements, probably due to the absorption of H$_2$
molecules in the pathway of the pyrolysis elemental analyzer (TCEA); this reduces the statistical precision of the δ$^2$H analyses.

## 3  Results and Discussion

### 3.1  Variations in the tree-ring cellulose δ$^{18}$O and δ$^2$H data





After constraining the exact dates for all the tree-rings by the cross-dating method using $\delta^{18}$O data, we compared all tree-ring cellulose $\delta^{18}$O and $\delta^2$H time-series (Fig. 3a and c).

Given that the average correlation coefficient (R-bar) amongst the different $\delta^{18}$O time-series is ca. 0.6–0.8, the Expressed Population Signal (EPS) (Wigley et al., 1984), defined as $N \times R / [1 + (N - 1) \times R]$ where the N and R values are the number of time-series and R-bar respectively, is >0.85 for almost all periods during the past 2,500 yr (Fig. 3b), indicating that the $\delta^{18}$O data in Fig. 3a are sufficient to establish accurate $\delta^{18}$O chronologies. However, there are distinct long-term age trends, in that the $\delta^{18}$O values gradually decrease as the trees mature, as also noted by Esper et al. (2010). These effects must be detrended in order to reconstruct long-term climate variations using the $\delta^{18}$O data. There are also significant offsets in absolute $\delta^{18}$O values for each year amongst different trees, probably reflecting differences in the sample location and magnitude of the age trend. Given the original locations of the trees cover a wide region from coastal to inland areas and low to high altitudes (Fig. 1), this would result in different precipitation $\delta^{18}$O values and relative humidity (Waseda and Nakai, 1983; Araguás-Araguás et al., 2000; Shu et al., 2005). This would account for the offset absolute $\delta^{18}$O values of tree-ring cellulose in different trees.

R-bar for the different $\delta^2$H time-series is ca. 0.2–0.4, resulting in lower EPS values (Fig. 3d), which partly reflects the analytical uncertainty due to the direct measurement of cellulose hydrogen including the OH-group and memory effects in the $\delta^2$H measurements. However, there are also distinct age trends, which are opposite to those exhibited by $\delta^{18}$O, in that $\delta^2$H values gradually increase as the trees mature.

## 3.2 Relationship between short- and long-periodicity $\delta^{18}$O and $\delta^2$H variations

To investigate the environmental causes of the age trends in tree-ring cellulose $\delta^{18}$O and $\delta^2$H data, we decomposed the $\delta^{18}$O and $\delta^2$H time-series into short- and long-periodicity variations, and compared these with tree-ring width variations in two trees that had experienced large changes in growth environments (Figs 4 and 5). Both trees (No. 65 and No. 49; Table 1) are Japanese cypress (*Chamaecyparis obtusa*) from Nagano Prefecture. In both cases, short-periodicity variations (i.e., <11 yr) in the tree-ring cellulose $\delta^{18}$O and $\delta^2$H data show a clear positive correlation (Figs 4c and 5c), indicating that both the $\delta^{18}$O and $\delta^2$H variations reflect a common climatological signal (i.e., precipitation isotope ratios and relative humidity) as reported in previous studies (e.g., An et al., 2014). However, the long-periodicity variations (i.e., >11 yr) in the tree-ring cellulose $\delta^{18}$O and $\delta^2$H data show negative correlations (Figs 4b and 5b), suggesting the influence of opposite age trends in Fig. 3. Figures 4b and 5b also show that the negative correlation is not simply due to the age trend, but is also related to changes in growth rate (i.e., tree-ring width).

Tree No. 65 germinated in the 18th century, grew rapidly in its juvenile stage, and shows a gradual decrease in growth rate during its first 100 yr (Fig. 4b). This is typical of 200–300 yr old trees of Japanese cypress, because the forests in central Japan were intensively logged in the 17–18th centuries at the beginning of the Edo era (17–19th century). Most old trees of Japanese cypress, including No. 65, germinated in open space after forest clearing where abundant light facilitated rapid growth in their





juvenile period (Totman, 1989). In addition, this tree experienced a sudden increase in growth rate (i.e., tree-ring width) at ca.
1950 CE, probably due to logging of neighboring trees after the end of World War II in 1945 CE (Fig. 4b). Tree No. 49
germinated in the 12[th] century, and survived the intense logging activity in the 17–18[th] century CE because it was left as a seed
tree. By the time of logging in the 17–18[th] centuries, the growth environment of the tree had improved step by step. We can
recognize three episodes of drastic growth rate increase of this tree at ca. 1600, 1710, and 1720 CE (Fig. 5b). After 1720 CE,
the growth rate decreased gradually along with that of neighboring (younger) trees.

Figures 4 and 5 show that $\delta^{18}O$ increased and $\delta^{2}H$ decreased, regardless of the tree age, corresponding to an increase in
tree-ring width (i.e., growth rate), in 1600, 1700, and 1720 CE for the No. 49 tree and 1950 CE for the No. 65 tree. Therefore,
the long-term opposite trends in the tree-ring cellulose $\delta^{18}O$ and $\delta^{2}H$ data are not a simple age trend. (Note that the long-term
$\delta^{18}O$ and $\delta^{2}H$ variations are not perfect mirror images because they are also affected by long-term climate variations in which
$\delta^{18}O$ and $\delta^{2}H$ are positively correlated.) Given the growth rates of trees that germinated in open spaces after logging or fires
gradually decline as the trees mature, the $\delta^{18}O$ and $\delta^{2}H$ values decrease and increase with time, respectively. However, the
growth rate not only decreases due to canopy closure, but also increases randomly due to human activities in the forest. As
such, it is impossible to apply the RCS or negative exponential curve proposed by Esper et al. (2010) for removing the apparent
age trend of tree-ring cellulose $\delta^{18}O$ in central Japan, where intense logging activity has been undertaken in forest environments
for a long time.

To date, the age trend in tree-ring cellulose $\delta^{18}O$ data has been explained by the relatively shallow root system of juvenile
trees and the evaporative enrichment of $^{18}O$ in soil water at the near-surface. However, this cannot explain the long-term
variations in tree-ring cellulose $\delta^{18}O$ data in this study (Figs 3–5), because the root depth cannot become shallower when the
growth rate increases and soil water evaporation must increase both $\delta^{18}O$ and $\delta^{2}H$. Figures 4b and 5b clearly show that the
long-term variations in tree-ring cellulose $\delta^{18}O$ and $\delta^{2}H$ are influenced by a physiological effect corresponding to long-term
changes in tree growth rate.

### 3.3 Mechanical model of cellulose $\delta^{18}O$ and $\delta^{2}H$

What mechanism can explain the physiological change in the $\delta^{18}O$ and $\delta^{2}H$ values corresponding to the change in growth
rate? Tree-ring cellulose $\delta^{18}O$ and $\delta^{2}H$ values can be expressed using the following equations, which are modified and
simplified from Roden et al. (2000):

$$\delta^{18}O_{cel} = f\,(\delta^{18}O_{rain} + \varepsilon_{ho}) + (1-f)[\delta^{18}O_{rain} + (\varepsilon_{eo} + \varepsilon_{ko})(1-h) + \varepsilon_{ao}], \tag{3}$$

$$\delta^{2}H_{cel} = f\,(\delta^{2}H_{rain} + \varepsilon_{hh}) + (1-f)[\delta^{2}H_{rain} + (\varepsilon_{eh} + \varepsilon_{kh})(1-h) + \varepsilon_{ah}], \tag{4}$$





where cel and rain indicate the tree-ring cellulose and precipitation, respectively; h is the relative humidity during photosynthesis; f is the proportion of oxygen atoms in carbohydrate exchanged with xylem water during post-photosynthetic
processes before cellulose synthesis, which is assumed to be equal to that of the hydrogen atoms; $\varepsilon_{eo}$ ($\varepsilon_{eh}$) and $\varepsilon_{ko}$ ($\varepsilon_{kh}$) are isotopic fractionation factors for oxygen (hydrogen) during equilibrium evaporation between water and water vapor and kinetic water vapor diffusion through leaf stomata, respectively; and $\varepsilon_{ao}$ ($\varepsilon_{ah}$) and $\varepsilon_{ho}$ ($\varepsilon_{hh}$) are isotopic fractionation factors for oxygen (hydrogen) between water and carbohydrate during photosynthesis in the leaf and the post-photosynthetic processes before cellulose synthesis, respectively. Values of $\varepsilon_{eo}$, $\varepsilon_{ko}$, $\varepsilon_{ao}$, and $\varepsilon_{ho}$ are +9, +29, +27, and +27, respectively, while $\varepsilon_{eh}$ and $\varepsilon_{kh}$ are
+80 and +25, respectively, and $\varepsilon_{ah}$ and $\varepsilon_{hh}$ are approximately –150 and +150, respectively (Roden et al., 2000).

Given that we can assume that all the isotopic fractionation factors are constant, $\delta^{18}O_{cel}$ and $\delta^2H_{cel}$ are determined only by climatological factors ($\delta^{18}O_{rain}$, $\delta^2H_{rain}$, and h) when f is constant. Because there are always positive correlations between $\delta^{18}O_{rain}$ and $\delta^2H_{rain}$ (Dansgaard, 1964) and any change of h produces changes in $\delta^{18}O_{cel}$ and $\delta^2H_{cel}$ in Eqs (3) and (4) in the same direction, we can expect that there are also positive correlations between $\delta^{18}O_{cel}$ and $\delta^2H_{cel}$ when f is constant. In fact, even in
trees that have negative correlations between $\delta^{18}O_{cel}$ and $\delta^2H_{cel}$ for the long-periodicity variations (Figs 4b and 5b), we observe clear positive correlations in the short-periodicity variations (Figs 4c and 5c), suggesting that f does not change on annual time-scales and the short-periodicity variations in $\delta^{18}O_{cel}$ and $\delta^2H_{cel}$ are controlled solely by climatological factors.

However, if f increases over the long-term due to some physiological reason, it can be inferred from Eqs (3) and (4) that $\delta^{18}O_{cel}$ decreases while $\delta^2H_{cel}$ increases. This is because, for oxygen, the magnitude of isotopic fractionation between water
and carbohydrate is the same (+27) for photosynthesis ($\varepsilon_{ao}$) and post-photosynthesis processes ($\varepsilon_{ho}$), and the increase in f reduces the effect of leaf water $^{18}O$ enrichment, ($\varepsilon_{Oe} + \varepsilon_{Ok}$)$(1 - h)$ in Eq. (3), resulting in lower $\delta^{18}O_{cel}$. However, for hydrogen, the magnitude of isotopic fractionation between water and carbohydrate during post-photosynthesis processes ($\varepsilon_{hh} = +150$) is significantly larger than that during photosynthesis ($\varepsilon_{ah} = -150$), such that the increase in f results in higher $\delta^2H_{cel}$. This causes long-term opposite trends in $\delta^{18}O_{cel}$ and $\delta^2H_{cel}$ data (Figs 3, 4b, and 5b), which can be interpreted as an increase in the rate of
post-photosynthetic isotopic exchange between carbohydrate and xylem water (f). Possible physiological mechanisms for this include an increase in the rate of utilization of stored carbohydrates for stem cellulose synthesis (Nabeshima et al., 2018), rather than using photosynthetic products directly for rapid tree growth during the juvenile period. This could also occur in the period following an abrupt improvement in the growth environment, due to logging of neighboring trees.

Although the long-periodicity variations in $\delta^{18}O_{cel}$ and $\delta^2H_{cel}$ are influenced by predominant physiological effects (Figs 4b
and 5b), this does not mean that the long-periodicity variations in $\delta^{18}O_{cel}$ and $\delta^2H_{cel}$ do not contain climatological components. In fact, climate varies at all time-scales, such that long-term variations in $\delta^{18}O_{cel}$ and $\delta^2H_{cel}$ inevitably include climatological components. It is therefore challenging to resolve the climatic signals from the physiological effects. In dendrochronological studies based on tree-ring width, RCS is used to separate and estimate climatological components in tree-ring width time-series (Esper et al., 2002; Grudd et al., 2002; Büntgen et al., 2005). However, it is difficult to create a regional standardized $\delta^{18}O_{cel}$



curve like the RCS for the samples analyzed in this study, because the physiological effects on $\delta^{18}O_{cel}$ are not solely an age

trend (Figs 4b and 5b), and also reflect past logging activity.

### 3.4  Classification of $\delta^{18}O$ and $\delta^{2}H$ variations into climatological and physiological components

To extract the climatological component of the variations in the tree-ring cellulose $\delta^{18}O$ data, we modified the model of

cellulose $\delta^{18}O$ and $\delta^{2}H$ in Eqs (3) and (4) into climatological and physiological components. Given there are four variables

($\delta^{18}O_{rain}$, $\delta^{2}H_{rain}$, h, and f) in Eqs (3) and (4), we first define their variations as follows:

$$\delta^{18}O_{rain} = \delta^{18}O_{rain(0)} + \Delta\delta^{18}O_{rain}, \tag{5}$$

$$\delta^{2}H_{rain} = \delta^{2}H_{rain(0)} + \Delta\delta^{2}H_{rain}, \tag{6}$$

$$h = h_{(0)} + \Delta h, \tag{7}$$

$$f = f_{(0)} + \Delta f, \tag{8}$$

where $\delta^{18}O_{rain(0)}$, $\delta^{2}H_{rain(0)}$, $h_{(0)}$, and $f_{(0)}$ are $\delta^{18}O_{rain}$, $\delta^{2}H_{rain}$, h, and f in a fixed year (0), respectively, and $\Delta\delta^{18}O_{rain}$, $\Delta\delta^{2}H_{rain}$, $\Delta h$,

and $\Delta f$ are deviations in $\delta^{18}O_{rain}$, $\delta^{2}H_{rain}$, h, and f from the fixed year (0) to an arbitrary year, respectively. By substituting Eqs

(5)–(8) into Eqs (3)–(4) and neglecting the second-order minor terms ($\Delta f\Delta h$, $\Delta f\Delta\delta^{18}O_{rain}$, and $\Delta f\Delta\delta^{2}H_{rain}$), Eqs (3) and (4) can

be rewritten as follows:

$$\begin{aligned} \delta^{18}O_{cel} = {} & f_{(0)}\left(\delta^{18}O_{rain(0)} + \varepsilon_{ho}\right) + \left(1 - f_{(0)}\right)\left[\delta^{18}O_{rain(0)} + (\varepsilon_{eo} + \varepsilon_{ko})\left(1 - h_{(0)}\right) + \varepsilon_{ao}\right] \\ & + \left[\Delta\delta^{18}O_{rain} - \Delta h(\varepsilon_{eo} + \varepsilon_{ko})\left(1 - f_{(0)}\right)\right] \\ & + \Delta f\left[\varepsilon_{ho} - \varepsilon_{ao} - (\varepsilon_{eo} + \varepsilon_{ko})\left(1 - h_{(0)}\right)\right], \end{aligned} \tag{9}$$

$$\begin{aligned} \delta^{2}H_{cel} = {} & f_{(0)}\left(\delta^{2}H_{rain(0)} + \varepsilon_{hh}\right) + \left(1 - f_{(0)}\right)\left[\delta^{2}H_{rain(0)} + (\varepsilon_{eh} + \varepsilon_{kh})\left(1 - h_{(0)}\right) + \varepsilon_{ah}\right] \\ & + \left[\Delta\delta^{2}H_{rain} - \Delta h(\varepsilon_{eh} + \varepsilon_{kh})\left(1 - f_{(0)}\right)\right] \\ & + \Delta f\left[\varepsilon_{hh} - \varepsilon_{ah} - (\varepsilon_{eh} + \varepsilon_{kh})\left(1 - h_{(0)}\right)\right]. \end{aligned} \tag{10}$$

We can now introduce new equations for the climatological and physiological components in the tree-ring cellulose $\delta^{18}O$

and $\delta^{2}H$ time-series, as follows:

$$\Delta\delta^{18}O_{cel(climate)} = \Delta\delta^{18}O_{rain} - \Delta h(\varepsilon_{eo} + \varepsilon_{ko})\left(1 - f_{(0)}\right), \tag{11}$$



$$\Delta\delta^2 H_{cel(climate)} = \Delta\delta^2 H_{rain} - \Delta h(\varepsilon_{eh} + \varepsilon_{kh})(1 - f_{(0)}), \tag{12}$$

$$\Delta\delta^{18}O_{cel(physiol)} = \Delta f\left[\varepsilon_{ho} - \varepsilon_{ao} - (\varepsilon_{eo} + \varepsilon_{ko})(1 - h_{(0)})\right], \tag{13}$$

$$\Delta\delta^2 H_{cel(physiol)} = \Delta f\left[\varepsilon_{hh} - \varepsilon_{ah} - (\varepsilon_{eh} + \varepsilon_{kh})(1 - h_{(0)})\right], \tag{14}$$

where $\Delta\delta^{18}O_{cel(climate)}$, $\Delta\delta^{18}O_{cel(physiol)}$, $\Delta\delta^2 H_{cel(climate)}$, and $\Delta\delta^2 H_{cel(physiol)}$ are variations in the tree-ring cellulose $\delta^{18}O$ and $\delta^2 H$ from the fixed year (0) with respect to an arbitrary year due to climatological and physiological factors, respectively. We can then reformulate $\delta^{18}O_{cel}$ and $\delta^2 H_{cel}$ as the sum of climatological and physiological components, as follows:

$$\delta^{18}O_{cel} = \delta^{18}O_{cel(0)} + \Delta\delta^{18}O_{cel(climate)} + \Delta\delta^{18}O_{cel(physiol)}, \tag{15}$$

$$\delta^2 H_{cel} = \delta^2 H_{cel(0)} + \Delta\delta^2 H_{cel(climate)} + \Delta\delta^2 H_{cel(physiol)}, \tag{16}$$

where $\delta^{18}O_{cel(0)}$ and $\delta^2 H_{cel(0)}$ are $\delta^{18}O_{cel}$ and $\delta^2 H_{cel}$ at the fixed year (0), respectively.

### 3.5 A method to extract the climatological component from cellulose $\delta^{18}O$

Here we propose a new method to calculate the climatological component in variations of the tree-ring cellulose $\delta^{18}O$ ($\Delta\delta^{18}O_{cel(climate)}$) by solving simultaneous equations consisting of Eqs (15)–(16), with two additional equations based on the relationship between $\Delta\delta^{18}O_{cel(climate)}$ and $\Delta\delta^2 H_{cel(climate)}$ and between $\Delta\delta^{18}O_{cel(physiol)}$ and $\Delta\delta^2 H_{cel(physiol)}$ (i.e., Eqs (17)–(18)). This is based on the theoretical and observational understanding that tree-ring cellulose $\delta^{18}O$ and $\delta^2 H$ data correlate positively and negatively due to climatological and physiological factors, respectively.

$$\Delta\delta^2 H_{cel(climate)} = A \times \Delta\delta^{18}O_{cel(climate)}, \tag{17}$$

$$\Delta\delta^2 H_{cel(physiol)} = -B \times \Delta\delta^{18}O_{cel(physiol)}, \tag{18}$$

where A and B are positive proportional coefficients, as shown in Eqs (19)–(20) and defined by Eqs (11)–(14).

$$A = \frac{\Delta\delta^2 H_{rain} - \Delta h(\varepsilon_{eh} + \varepsilon_{kh})(1 - f_{(0)})}{\Delta\delta^{18}O_{rain} - \Delta h(\varepsilon_{eo} + \varepsilon_{ko})(1 - f_{(0)})}, \tag{19}$$

$$B = -\frac{\varepsilon_{hh} - \varepsilon_{ah} - (\varepsilon_{eh} + \varepsilon_{kh})(1 - h_{(0)})}{\varepsilon_{ho} - \varepsilon_{ao} - (\varepsilon_{eo} + \varepsilon_{ko})(1 - h_{(0)})}, \tag{20}$$





If A and B are constant, we can solve the simultaneous equations (15)–(18) in order to cancel out the physiological component ($\Delta\delta^{18}O_{cel(physiol)}$) and extract the climatological component ($\Delta\delta^{18}O_{cel(climate)}$), as follows:

$$\Delta\delta^{18}O_{cel(climate)} = \frac{\delta^2H_{cel} + B \times \delta^{18}O_{cel}}{A + B} - \frac{\delta^2H_{cel(0)} + B \times \delta^{18}O_{cel(0)}}{A + B},\qquad(21)$$

In fact, A might not be constant because there are three variables ($\Delta\delta^{18}O_{rain}$, $\Delta\delta^2H_{rain}$, and $\Delta h$) in Eq. (19). $\Delta\delta^{18}O_{rain}$ and $\Delta\delta^2H_{rain}$ may change somewhat independently from each other over a wide spatiotemporal range, and the relationship between $\Delta\delta^{18}O_{rain}$ ($\Delta\delta^2H_{rain}$) and $\Delta h$ may not be simple in Eq. (19). However, the significant positive correlations in the short-periodicity variations between $\delta^{18}O_{cel}$ and $\delta^2H_{cel}$ (Figs 4c and 5c) suggest there may be an empirical constant value for A in a restricted
study area such as central Japan. In contrast to A, B may be nearly constant because there are no variables in Eq. (20).

Although the assumption that A and B are constant might not be valid, this assumption is needed to explicitly calculate $\Delta\delta^{18}O_{cel(climate)}$ using Eq. (21). Hence, we tentatively assumed that A and B are constant, and calculated $\Delta\delta^{18}O_{cel(climate)}$ over the past 2,600 yr using Eq. (21). We then verify this by comparison with numerous local, regional, and global meteorological, historical, and paleoclimatological records of past summer climate over various time-scales.

**3.6 Method to determine the proportional coefficients A and B**

To utilize Eq. (21), we need to determine the proportional coefficients A and B in Eqs (17)–(20). In principle, there are two ways to determine A and B: a theoretical approach based on Eqs (19)–(20), and an empirical approach based on Eqs (17)–(18).

We first consider the feasibility of the theoretical approach. It is not easy to determine A using Eq. (19), because there are
three variables ($\Delta\delta^{18}O_{rain}$, $\Delta\delta^2H_{rain}$, and $\Delta h$). If the rainwater isotope ratios do not change (both $\Delta\delta^{18}O_{rain}$ and $\Delta\delta^2H_{rain} = 0$), then A is equal to $(\varepsilon_{eh} + \varepsilon_{kh})/(\varepsilon_{eo} + \varepsilon_{ko})$, which is $(80 + 25)/(9 + 29) = 2.76$ (Roden and Ehleringer, 2000). However, if relative humidity does not change ($\Delta h = 0$), $A = \Delta\delta^2H_{rain}/\Delta\delta^{18}O_{rain}$, which is equal to 8 if the meteoric water line is followed (Dansgaard, 1964). Given the wide range of potential A values from 2.76 to 8, we cannot easily theoretically define A. On the other hand, B may be easier to determine theoretically, because there are no variables in Eq. (20). For example, if the relative humidity
$h_{(0)}$ is 0.5, B = 13 because $\varepsilon_{eo}$, $\varepsilon_{ko}$, $\varepsilon_{ao}$, $\varepsilon_{ho}$, $\varepsilon_{eh}$, $\varepsilon_{kh}$, $\varepsilon_{ah}$, and $\varepsilon_{hh}$ are +9, +29, +27, +27, +80, +25, ca. –150, and +150, respectively (Roden et al., 2000). However, in fact, it is not easy to fix $h_{(0)}$ because the relative humidity typically shows large diurnal variations, and the timing of $\delta^{18}O$ incorporation into leaf carbohydrate is unknown. Moreover, the $\varepsilon_{ah}$ and $\varepsilon_{hh}$ values of –150 and +150 are just approximations. For example, while Yakir and DeNiro (1990) obtained values of –171 and +158 for $\varepsilon_{ah}$ and $\varepsilon_{hh}$, respectively, Estep and Hoering (1981) obtained values of –100 to –120 for $\varepsilon_{ah}$ and Luo and Sternberg (1992) reported
values of +144 to +166 for $\varepsilon_{hh}$.





Therefore, we used the empirical approach for estimating A and B. The A value in Eq. (17) can be estimated from the relationship between $\delta^{18}O_{cel}$ and $\delta^2H_{cel}$ in the short-periodicity variations, because they are positively correlated due to the climate variations shown in Figs 4c and 5c. In the case of the B value in Eq. (18), the situation is more complex. Although there are apparent negative correlations between $\delta^{18}O_{cel}$ and $\delta^2H_{cel}$ in the long-periodicity variations (Figs 4b and 5b), these

are also affected by long-term climate signals, such that it is difficult to resolve the physiological component. Here we propose a practical method to determine the B value. After fixing A, we first set various B values in order to calculate the temporal variations in the climatological component of the tree-ring cellulose $\delta^{18}O$ data ($\Delta\delta^{18}O_{cel(climate)}$) for all 67 tree samples, by substituting individual $\delta^{18}O_{cel}$ and $\delta^2H_{cel}$ time-series in Fig. 3a and c into Eq. (21). We then combine the $\Delta\delta^{18}O_{cel(climate)}$ time-series of all 67 trees calculated from each B value using the method described below (Sect. 3.7). Finally, we compare the

overall trend of the combined time-series of $\Delta\delta^{18}O_{cel(climate)}$ data with that of a simple average of all 67 raw $\delta^{18}O_{cel}$ time-series in Fig. 3a, in order to find the B value that produced the best match between the two time-series. This procedure assumes that, although individual trees possess different age trends and level offsets, utilizing all data from all trees cancels out these effects and allows climate variations to be discerned, at least in terms of the overall trend.

### 3.7 Method to combine individual tree-ring time-series with large level offsets

To reconstruct multi-millennial variations in the climatological component of the tree-ring cellulose $\delta^{18}O$ data ($\Delta\delta^{18}O_{cel(climate)}$), we must combine individual time-series from different trees with variable level offsets due to different sample locations. As such, we cannot simply average all data for individual trees, because this produces steps in the composite record at both ends of the time-series of individual trees. Numerous procedures have been proposed to combine tree-ring time-series (e.g., Hangartner et al., 2012); here we propose a new iterative calculation method (Fig. 6).

Firstly, we simply average all the individual time-series to make a preliminary combined time-series. Secondly, we offset each individual time-series up or down, retaining their original patterns of temporal variations, to the position where the average of the individual time-series becomes equal to that of the preliminary combined time-series for the corresponding period of the individual tree. Thirdly, we average all the offsetted individual time-series to make a refined combined time-series. We iterate this procedure until the average of the individual time-series becomes equal to that of the combined time-

series for the corresponding period, without any further offsetting of the individual time-series. This method assumes that the tree-ring absolute isotope ratios of individual trees are not important, because they depend on the sample locations, but that the temporal variations are well correlated amongst different trees due to a common regional climate signal.

### 3.8 Procedure to calculate the climatological component in tree-ring $\delta^{18}O$ ($\Delta\delta^{18}O_{cel(climate)}$) data

Figure 7a shows how the combined time-series for $\Delta\delta^{18}O_{cel(climate)}$ is constructed. Firstly, we determine A from the short-

periodicity variations between $\delta^{18}O_{cel}$ and $\delta^2H_{cel}$ in individual trees. Secondly, we use $\delta^{18}O_{cel}$ and $\delta^2H_{cel}$ to calculate





$\Delta\delta^{18}O_{cel(climate)}$ for individual trees using various B values in Eq. (21). Thirdly, we combine the $\Delta\delta^{18}O_{cel(climate)}$ time-series of all trees using various B values. Fourthly, we select the best B value by comparison of the overall trends in the combined $\Delta\delta^{18}O_{cel(climate)}$ and average $\delta^{18}O_{cel}$ data. Finally, in order to minimize the influence of the lower precision $\delta^2H$ measurements (Fig. 3d), we reintegrate the long-periodicity variations in $\Delta\delta^{18}O_{cel(climate)}$ and short-periodicity variations in $\delta^{18}O_{cel}$ for all

individual trees at an adequate threshold periodicity and combine all individual data again, because the physiological factors do not appear to affect the short-periodicity $\delta^{18}O_{cel}$ variation. In this procedure (Fig. 7a), the iterative calculation used to combine many tree-ring time-series in the third step is the most time-consuming because various B values are tested.

  Although A can be determined independently for each tree, we can only obtain one B value for all the trees. In a practical sense, it is not meaningful to determine A separately for each tree, because all the trees were collected within central Japan

(Fig. 1). If we assume that A and B are unique for all trees in this study, we can simplify the procedure as shown in Fig. 7b to reduce the time required for the iterative calculation. Considering that all the calculations used to combine and integrate the time-series in Fig. 7 are linear, we can change the order of calculation between combination and integration. In fact, if we use common A and B values, the resultant combined time-series for $\Delta\delta^{18}O_{cel(climate)}$ does not change when using the two procedures (Fig. 7a and b). In Fig. 7b, we first combine the $\delta^{18}O_{cel}$ and $\delta^2H_{cel}$ time-series of all trees in Fig. 3, using the method shown in

Fig. 6. Secondly, we determine A by investigating the relationship between short-periodicity variations of $\delta^{18}O_{cel}$ and $\delta^2H_{cel}$ in their combined time-series. Thirdly, we integrate the combined $\delta^{18}O_{cel}$ and $\delta^2H_{cel}$ records to show temporal variations in $\Delta\delta^{18}O_{cel(climate)}$ using various B values. Fourthly, we select the best B value by comparison of the overall trends of the $\Delta\delta^{18}O_{cel(climate)}$ time-series and average $\delta^{18}O_{cel}$. Finally, we reintegrate the long-periodicity variations in $\Delta\delta^{18}O_{cel(climate)}$ and short-periodicity variations in combined $\delta^{18}O_{cel}$ at an adequate threshold periodicity to minimize the influence of the lower precision

$\delta^2H$ data.

### 3.9 Determination of the climatological and physiological proportional coefficients (A and B)

  We combined all the time-series for $\delta^{18}O_{cel}$ and $\delta^2H_{cel}$ shown in Fig. 3 using the method illustrated in Fig. 6. The final combined $\delta^{18}O_{cel}$ and $\delta^2H_{cel}$ time-series after 1000 iterations are shown in Fig. 8. The long-term variations in the combined $\delta^{18}O_{cel}$ and $\delta^2H_{cel}$ time-series obviously reflect the accumulated age trends, in which the $\delta^{18}O_{cel}$ and $\delta^2H_{cel}$ tend to decrease and

increase over a long time-scale, respectively.

  We found that the climatological proportional coefficient A in Eq. (17) can be set to 5, because there are positive correlations in the short-periodicity variations of the combined $\delta^2H_{cel}$ and $\delta^{18}O_{cel}$ records with a slope of ca. 5 (Fig. 9), irrespective of the threshold year for extracting the short-periodicity variations (i.e., 5, 11, or 21 yr). A value of 5 is within the theoretical range of A of 2.76 to 8 obtained from Eq. (19). We used the $\delta^2H_{cel}$ data directly for the calculation of A, although

the amplitude of variations in $\delta^2H_{cel}$ is reduced to 70% of that of the original cellulose, such that a value of 5 is equivalent to 7 in the theoretical range.



To determine the most appropriate value for the physiological proportional coefficient B in Eq. (18), we integrated the combined $\delta^{18}O_{cel}$ and $\delta^2H_{cel}$ time-series in Fig. 10 using Eq. (21), in order to calculate $\Delta\delta^{18}O_{cel(climate)}$ with B values of 3, 5, 7, and 9, and an A value 5. We then examined the overall trend in $\Delta\delta^{18}O_{cel(climate)}$ and found that it became equal to the overall

trend of the average $\delta^{18}O_{cel}$ (Fig. 3a), when B = 5.4 (Fig. 10). Given the 70% amplitude reduction of $\Delta\delta^2H_{cel}$, 5.4 is equivalent to 7.7 from the theoretical estimation. If we assume that $\varepsilon_{eo}$, $\varepsilon_{ko}$, $\varepsilon_{ao}$, $\varepsilon_{ho}$, $\varepsilon_{eh}$, $\varepsilon_{kh}$, $\varepsilon_{ah}$, and $\varepsilon_{hh}$ are +9, +29, +27, +27, +80, +25, –150, and +150, respectively, then B = 7.7 means that $h_{(0)}$ is 0.25 in Eq. (20). Given that 0.25 is too low for the relative humidity in central Japan, the values of –150 and +150 for $\varepsilon_{ah}$ and $\varepsilon_{hh}$, respectively, may be overestimated. However, because the overall trend of the integrated $\Delta\delta^{18}O_{cel(climate)}$ using B = 5.4 is equal to that of the average of raw $\delta^{18}O_{cel}$ values, we use this value of B

hereafter.

## 3.10 Calculation of temporal variations in the climatological component ($\Delta\delta^{18}O_{cel(climate)}$)

We calculated the temporal variation in the climatological component ($\Delta\delta^{18}O_{cel(climate)}$) of tree-ring cellulose $\delta^{18}O$ (Fig. 11a) by using A = 5 and B = 5.4, and the smoothly combined $\delta^{18}O_{cel}$ and $\delta^2H_{cel}$ time-series in Fig. 8 and Eq. (21). We used the temporal average of $(\delta^2H_{cel} + B\,\delta^{18}O_{cel})/(A + B)$ during the 30 yr from 1961 to 1990 as $(\delta^2H_{cel(0)} + B\,\delta^{18}O_{cel(0)})/(A + B)$ in Eq.

360    (21).

The $\Delta\delta^{18}O_{cel(climate)}$ variations shown in Fig. 11a must be influenced by the low quality (low R-bar and EPS values) of the original $\delta^2H_{cel}$ time-series (Fig. 3d), because $\Delta\delta^{18}O_{cel(climate)}$ gives equal weighting to $\delta^{18}O_{cel}$ and $\delta^2H_{cel}$. In fact, when we calculated EPS values for all individual $\Delta\delta^{18}O_{cel(climate)}$ time-series from Eq. (21), we found that the EPS values are significantly lower, reflecting the low EPS value of the original $\delta^2H_{cel}$ time-series (Fig. 11b). The purpose of introducing the $\delta^2H_{cel}$ signal

into the $\delta^{18}O_{cel}$ time-series was to remove the physiological effects from the $\delta^{18}O_{cel}$ time-series, but the short-periodicity variations in the $\delta^{18}O_{cel}$ time-series do not originally contain physiological effects, so it is not necessary to integrate the $\delta^2H_{cel}$ signals into the short-periodicity components of the $\delta^{18}O_{cel}$ time-series. Therefore, at two threshold periodicities (21 and 51 yr), we tentatively separated the long-periodicity component (21- and 51-yr running means) from the integrated $\Delta\delta^{18}O_{cel(climate)}$ time-series in Fig. 11a and short-periodicity component (deviations from 21- and 51-yr running means) from the original

combined $\delta^{18}O_{cel}$ in Fig. 8, and reintegrated these into a new time-series for $\Delta\delta^{18}O_{cel(climate)}$ to remove the influence of the low quality $\delta^2H_{cel}$ data from the short-periodicity component. We also applied this reintegration procedure between the long ($\Delta\delta^{18}O_{cel(climate)}$) and short ($\delta^{18}O_{cel}$) periodicity components to the data for individual trees, and calculated the EPS values for the reintegrated $\Delta\delta^{18}O_{cel(climate)}$ datasets (Fig. 11b). The resultant EPS values were >0.85, and much higher than the original $\Delta\delta^{18}O_{cel(climate)}$ for almost all periods during last 2,500 yr when either 21 or 51 yr were used as the thresholds. This suggests

that the reintegration procedure ensures the reliability of the datasets without the influence of the low-quality $\delta^2H_{cel}$ data.

However, this reintegration procedure may recover potential physiological effects with intermediate periodicities of less than 21 and 51 yr. Hence, we compared the long-term variations (periodicity > 11 yr) of the two reintegrated $\Delta\delta^{18}O_{cel(climate)}$



time-series using 21 and 51 yr as the thresholds and the original $\Delta\delta^{18}O_{cel(climate)}$ time-series (Fig. 11a) in Fig. 11c to investigate whether there are significant differences. Each of the three time-series shown in Fig. 11c almost coincide over all time-scales, indicating there are no significant physiological effects with a periodicity between 11 and 51 yr. However, we used 21 yr as the threshold for the reintegration of the long- and short-periodicity components, in order to robustly remove the influence of physiological effects. It does not result in a lower quality reintegrated $\Delta\delta^{18}O_{cel(climate)}$ record, given the nearly equal EPS values using the 21- and 51-yr thresholds in Fig. 11b. We utilized the reintegrated $\Delta\delta^{18}O_{cel(climate)}$ time-series between the long-periodicity (>21 yr) domain of $\Delta\delta^{18}O_{cel(climate)}$ in Fig. 11a and short-periodicity (<21 yr) domain of $\delta^{18}O_{cel}$ in Fig. 8 as the final time-series of the climatological component in the tree-ring cellulose $\delta^{18}O$ data (Fig. 12).

In contrast to the combined $\delta^2H_{cel}$ record (Fig. 8), the multi-centennial variation in the combined $\delta^{18}O_{cel}$ record (Fig. 8) does not appear to be very similar to that of the climatological component $\Delta\delta^{18}O_{cel(climate)}$ (Fig. 12). This is partly because there is an apparent age trend in the combined $\delta^{18}O_{cel}$ record, which overlaps the multi-millennial climatological decrease in Fig. 10. This means the multi-centennial variations are ambiguous in the combined $\delta^{18}O_{cel}$ record (Fig. 8). However, there may be an anthropogenic explanation for this, whereby in wetter and cooler periods, the number of trees in Japanese forests might have decreased due to logging for fuel. Rapid tree growth in the resultant more open and lighter forest would have increased $\delta^{18}O_{cel}$ and decreased $\delta^2H_{cel}$ values due to physiological effects. The wetter and cooler climate may have also lowered $\delta^{18}O$ and $\delta^2H$ in leaf water. The combined effects of climate variations and anthropogenic factors on Japanese forests might have reduced and enhanced the multi-centennial variations in $\delta^{18}O_{cel}$ and $\delta^2H_{cel}$, respectively (Fig. 8). However, we can robustly extract the climatological component $\Delta\delta^{18}O_{cel(climate)}$ independently of the local forest history by integrating $\delta^{18}O_{cel}$ and $\delta^2H_{cel}$ data. This is the most important paleoclimatological innovation of our study.

### 3.11 Comparison of $\Delta\delta^{18}O_{cel(climate)}$ with other summer climate records

In dendroclimatological studies focused on inter-annual variability, statistical methods to calibrate and verify the relationship between tree-ring data and instrumental meteorological observations have been well established. However, there is no commonly accepted statistical method to validate the reliability of long (i.e., centennial or millennial) periodicity climate reconstructions. In the case of low-frequency data recovered from speleothems, ice cores, and sediments, climate reconstructions are typically not based on correlations with meteorological observations. These reconstructions are verified by different methods, such as mechanical models based on the relationship between oxygen isotope ratios and environmental factors, empirical knowledge of the relationship between pollen assemblages and climate, and experimental studies between biomarker compositions and water temperature. The reconstruction of $\Delta\delta^{18}O_{cel(climate)}$ variations is principally based on the mechanical model developed in Eqs (3)–(21), but it is necessary to validate our results by comparison with other past summer climate records.

Figure 13 shows the sensitivity of $\Delta\delta^{18}O_{cel(climate)}$ to local monthly mean temperature, mean relative humidity, and precipitation during the period from 1901–2005 at Kyoto, Nagoya, and Iida in central Japan. $\Delta\delta^{18}O_{cel(climate)}$ shows significant





negative correlations with precipitation and relative humidity, and a positive correlation with temperature during summer (Fig.
13d–f), when annual precipitation is at its maximum (Fig. 13a–c), as demonstrated by previous studies of monsoonal Asia (Li
et al., 2015; Liu et al., 2017; Pumijumnong et al., 2019; Sano et al., 2012, 2013, 2017; Seo et al., 2019; Xu et al., 2013, 2018,
2019). Spatial correlations of $\Delta\delta^{18}O_{cel(climate)}$ with June–July precipitation in East Asia (Fig. 14) indicate that $\Delta\delta^{18}O_{cel(climate)}$ in
central Japan reflects precipitation in an extended region from the lower reaches of the Yangtze River in China to southern

Honshu in Japan, corresponding to the Baiu/Meiyu front, which is an early summer stagnant rain belt characteristic of the East
Asian summer monsoon (Fig. 14a). $\Delta\delta^{18}O_{cel(climate)}$ has a significant positive correlation with June–July mean temperature
across a wide region of Japan, Korea, and China, suggesting that $\Delta\delta^{18}O_{cel(climate)}$ may be a good proxy for the East Asia summer
monsoon (Fig. 14c).

The negative correlations between $\Delta\delta^{18}O_{cel(climate)}$ and relative humidity and precipitation can be explained by the direct

negative relationship between $\Delta\delta^{18}O_{cel(climate)}$ and relative humidity in Eq. (11) and the amount effect, whereby $\delta^{18}O_{rain}$ becomes
lower when precipitation increases (Dansgaard, 1964; Araguás-Araguás et al., 1998). The highest correlation (<0.6) area of
June–July precipitation is located just to the south of the sample sites (Fig. 14b). This is because, in the summer season, water
vapor usually comes from the south (Pacific Ocean) to the sample sites and $\delta^{18}O_{rain}$ becomes lower when heavy rainfall occurs
just before arrival of the air mass carrying the water vapor. The positive correlation between $\Delta\delta^{18}O_{cel(climate)}$ and temperature

must be caused by the meteorologically reverse relationship between summer precipitation and temperature in humid
monsoonal Asia, including Japan. In fact, the center of the highest correlation area of June–July mean temperature is located
slightly to the west of the sample site (Fig. 14c). As such, when the temperature in western Japan is high and it is characterized
by high pressure, the wind blows from the north to the sample site resulting in dry conditions and low rainfall.

Given the relationship between $\Delta\delta^{18}O_{cel(climate)}$ and modern meteorological observations evident in Figs 13 and 14, we

compared $\Delta\delta^{18}O_{cel(climate)}$ in central Japan with long-term historical and paleoclimatological records of summer climate before
the 19[th] century. During the Edo era (1603–1868 CE), people wrote numerous diaries throughout Japan in which daily weather
conditions were routinely described. Mizukoshi (1993) compiled many diary weather descriptions for central Japan and
reconstructed inter-annual variations in early summer precipitation for Osaka since 1692 CE (Fig. 15). The diary-based (1692–
1882) and instrumentally observed (1883–1990) precipitation reconstructions for Osaka are negatively correlated with

$\Delta\delta^{18}O_{cel(climate)}$ in central Japan, not only at an inter-annual time-scale, but also at a multi-decadal time-scale, indicating that
$\Delta\delta^{18}O_{cel(climate)}$ records long-term variations in summer climate.

During the Medieval Period from the 11[th] to 16[th] centuries in Japan, there were numerous extreme meteorological events
(Fujiki, 2007). We used flood- and drought-related disaster records during the summer season (June–August) to construct an
index of the "flood disaster ratio" that is the proportion of flood-related document numbers to the total flood- and drought-

related document numbers. Given the total document numbers are scarcer in the older period, we calculated an 11-yr running
mean of the "flood disaster ratio" and compared it with $\Delta\delta^{18}O_{cel(climate)}$ (Fig. 16). In the 10[th] and 11[th] centuries during the
Medieval Climate Anomaly, there were numerous droughts in Japan, corresponding to the highest values of $\Delta\delta^{18}O_{cel(climate)}$.



After the 11$^{th}$ century, both the documentary records and $\Delta\delta^{18}O_{cel(climate)}$ values demonstrate that climate became progressively wetter towards the Edo era in the 17$^{th}$ century.

Although we could not find historical records of extreme climate events prior to the 10$^{th}$ century CE in Japan, Sakaguchi (1983, 1989) reconstructed long-term summer temperature variations using the pollen percentage of a cold region pine (*Pinus pumila*) in the Ozegahara peatland of east Japan (Fig. 17). Although the pollen data after the 3$^{rd}$ century CE are not very reliable due to the lower sedimentation rate and human disturbance, the high sedimentation rate before the 2$^{nd}$ century CE enabled us to compare it with $\Delta\delta^{18}O_{cel(climate)}$. Both the pollen and $\Delta\delta^{18}O_{cel(climate)}$ records show similar variations from the 6$^{th}$ century BCE

to 2$^{nd}$ century CE, indicating a warmer and drier climate from the 5$^{th}$ to 2$^{nd}$ century BCE and a cooler and wetter climate after the 1$^{st}$ century BCE (Fig. 17). After the 3$^{rd}$ century CE, both datasets show similar millennial variations, although the temporal resolution of the pollen data is not high.

    The climatological component of the variations in tree-ring cellulose $\delta^{18}O$ ($\Delta\delta^{18}O_{cel(climate)}$) correlates well with meteorological, historic, and vegetation data over various time-scales in central Japan, and also shows similar long-term

patterns as paleoclimatological global temperature and East Asian precipitation data. Figure 18a shows that $\Delta\delta^{18}O_{cel(climate)}$ exhibits almost identical variations as air temperature reconstructions for land areas in the Northern Hemisphere (Mann et al., 2008), except for the period of the Medieval Climate Anomaly. Variations in summer precipitation reconstructed from diatom assemblages in lake sediments in Taiwan (Wang et al., 2013) are also similar to the variations in $\Delta\delta^{18}O_{cel(climate)}$ in central Japan (Fig. 18b). Two time-series of carbonate $\delta^{18}O$ values in speleothems from northwest China (Zhang et al., 2008; Tan et al.,

2010) also match the variations in $\Delta\delta^{18}O_{cel(climate)}$ (Fig. 18c). Note that the directions of the y-axis are reversed between the speleothem and $\Delta\delta^{18}O_{cel(climate)}$ data, reflecting the meridional disparity of precipitation patterns in East Asia (Fig. 14a), as demonstrated by Liu et al. (2014) and Chen et al. (2015).

    The climatological component of the variations in the tree-ring cellulose $\delta^{18}O$ data ($\Delta\delta^{18}O_{cel(climate)}$) in central Japan (Fig. 12) correspond well over various time-scales with summer precipitation and temperature records in central Japan, which have

been derived from various meteorological, historical, and paleo-vegetation archives (Figs 13–17). This indicates that $\Delta\delta^{18}O_{cel(climate)}$ is a reliable proxy of summer climate, such as the activity of the East Asian summer monsoon. Multi-centennial and millennial $\Delta\delta^{18}O_{cel(climate)}$ variations are similar to those of paleoclimatological reconstructions of global temperatures and East Asian precipitation (Fig. 18), indicating a drier climate during the Medieval Climate Anomaly and wetter climate during the Little Ice Age in central Japan. As such, $\Delta\delta^{18}O_{cel(climate)}$ can be utilized in climatological, historical, and archeological studies.

**4 Conclusions**

    We constructed a statistically reliable multi-millennial tree-ring dataset of cellulose $\delta^{18}O$ in central Japan by analysing tree-rings of 67 trees without using a pooling method. We found that there are distinct age trends in the $\delta^{18}O$ time-series. By comparison with the $\delta^{2}H$ time-series, we showed that the age trend in $\delta^{18}O$ is caused by an increase in the degree of post-



photosynthesis isotopic exchange with xylem water before cellulose synthesis as the trees mature. Because the physiological
conditions of the post-photosynthesis isotopic exchange are not simply controlled by tree age, but also related to the tree growth
environment randomly influenced by human activity, it was not possible to remove the age trend by application of the negative
exponential curve or RCS.

Given that tree-ring cellulose $\delta^{18}O$ and $\delta^2H$ are correlated positively and negatively due to climatological and physiological
factors, respectively, we formulated simultaneous equations for the climatological and physiological components of the tree-
ring cellulose $\delta^{18}O$ and $\delta^2H$ data. We solved these equations to cancel out the physiological effects and established a multi-
millennial record of the climatological component of tree-ring cellulose $\delta^{18}O$ ($\Delta\delta^{18}O_{cel(climate)}$). The $\Delta\delta^{18}O_{cel(climate)}$ time-series
is well correlated with local, regional, and global variations in summer climate reconstructed by instrumental, historical, and
paleoclimatological methods. This suggests that $\Delta\delta^{18}O_{cel(climate)}$ records summer climate variations in central Japan during the
past 2,600 yr on annual to millennial time-scales.

However, further research is needed to make $\Delta\delta^{18}O_{cel(climate)}$ a more reliable index of summer climate. Firstly, the analytical
precision of the tree-ring cellulose $\delta^2H$ measurements needs to be improved. In order to minimize the negative influence of
the exchangeable OH-group hydrogen, it is necessary to fix the $\delta^2H$ in the OH-group (Filot et al., 2006). The memory effect
of $H_2$ molecules in the pyrolysis elemental analyzer also needs to be reduced. Secondly, more global tree-ring cellulose $\delta^2H$
data need to be acquired to expand the $\delta^{18}O$ and $\delta^2H$ datasets (and $\Delta\delta^{18}O_{cel(climate)}$ time-series) for different regions and tree
species and validate its usefulness. Thirdly, it remains challenging to calibrate $\Delta\delta^{18}O_{cel(climate)}$ variations, because there are few
long-term datasets of meteorological observations.

Although $\Delta\delta^{18}O_{cel(climate)}$ is a promising proxy for summer climate that can remove the age trend, previous $\delta^{18}O$-based
dendrochronological reconstructions of past summer climate may still be robust. Most tree-ring $\delta^{18}O$ studies utilized samples
collected from natural forests where human disturbance was insignificant in comparison with forests in central Japan. In
contrast to the conifer trees studied here and reported by Esper et al. (2010), hardwoods may be intrinsically free from long-
term age trends (Duffy et al., 2017). Therefore, in most isotopic dendrochronological studies, the cellulose $\delta^2H$ data will be a
supplementary index to ensure there are no significant age trends (An et al., 2014). However, when such studies are based on
a small number of conifer woods collected from archeological artifacts and/or architectural material, and where their growth
environments may have been disturbed by human activities, the simultaneous measurement of $\delta^2H$ with $\delta^{18}O$ allows
$\Delta\delta^{18}O_{cel(climate)}$ variations to be used as a proxy of past climate.

*Data availability.*   The data obtained in this research are available at https://www.ncdc.noaa.gov/paleo/study/28832.

*Author contributions.* TN designed the research and wrote the paper; TN, MS, KS, TM, MS, HO, NH, NN, MY collected tree
ring samples; TN, MS, KS, ZL, CX, KO extracted cellulose from tree rings; TN, MS, ZL, CX, KO, AT measured isotope ratios



of tree-ring cellulose; TN, MS, YS analyzed isotopic data mathematically; All authors discussed the results and provided input
to the manuscript.

*Competing interests.* The authors declare that they have no conflict of interest.

*Acknowledgement*s. We thank K. Oishi, M. Okabe, and A. Ishida for their support with the analysis of the tree-ring cellulose
isotopic ratios, and J. Akatsuka and M. Hakozaki for their assistance in collecting important tree-ring samples. We are grateful
to the Nagoya University Museum, Iida-Kamisato Archaeological Museum, and Akazawa Forest Museum in the Kiso Forest
Management Office for providing samples. This research was performed in the Research Institute for Humanity and Nature
(RIHN: a constituent member of NIHU) as Project No. 14200077 (Historical Climate Adaptation Project), and was supported
by Grants-in-Aid for Scientific Research from the Japanese Society for the Promotion of Science (Grants 23242047, 26244049,
and 17H06118).

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

**Table and Figures**



**Table 1.** List of samples including type of sample, tree species, age range, and place of collection.

| No. of sample | Type of wood *1 | Tree species | Start year*2 | End year*2 | Location of municipalities (Prefecture) | Latitude & Longitude (Local government*3) |
|---|---|---|---|---|---|---|
| 1 | A | *Chamaecyparis obtusa* | -612 | -354 | Nagoya city(Aichi) | 35.18N, 136.91E |
| 2 | N | *Chamaecyparis obtusa* | -543 | -194 | Iida city (Nagano) | 35.51N, 137.82E |
| 3 | A | *Chamaecyparis obtusa* | -537 | -434 | Nagoya city(Aichi) | 35.18N, 136.91E |
| 4 | A | *Chamaecyparis obtusa* | -530 | -288 | Nagoya city(Aichi) | 35.18N, 136.91E |
| 5 | A | *Chamaecyparis obtusa* | -465 | -340 | Nagoya city(Aichi) | 35.18N, 136.91E |
| 6 | A | *Chamaecyparis obtusa* | -380 | -279 | Nagoya city(Aichi) | 35.18N, 136.91E |
| 7 | A | *Cryptomeria japonica* | -348 | -238 | Nagoya city(Aichi) | 35.18N, 136.91E |
| 8 | A | *Chamaecyparis pisifera* | -276 | -158 | Nagoya city(Aichi) | 35.18N, 136.91E |
| 9 | N | *Chamaecyparis obtusa* | -267 | 71 | Iida city (Nagano) | 35.51N, 137.82E |
| 10 | A | *Chamaecyparis obtusa* | -253 | -183 | Nagoya city(Aichi) | 35.18N, 136.91E |
| 11 | A | *Sciadopitys verticillata* | -225 | -70 | Kasugai city (Aichi) | 35.25N, 136.97E |
| 12 | A | *Sciadopitys verticillata* | -216 | -66 | Kasugai city (Aichi) | 35.25N, 136.97E |
| 13 | N | *Chamaecyparis obtusa* | -177 | 207 | Koga city (Shiga) | 34.97N, 136.17E |
| 14 | A | *Chamaecyparis obtusa* | -167 | -56 | Nagoya city(Aichi) | 35.18N, 136.91E |
| 15 | A | *Sciadopitys verticillata* | -167 | -57 | Kasugai city (Aichi) | 35.25N, 136.97E |
| 16 | A | *Chamaecyparis obtusa* | -83 | 32 | Nagoya city(Aichi) | 35.18N, 136.91E |
| 17 | N | *Chamaecyparis obtusa* | -65 | 169 | Iida city (Nagano) | 35.51N, 137.82E |
| 18 | N | *Chamaecyparis obtusa* | -20 | 442 | Koga city (Shiga) | 34.97N, 136.17E |
| 19 | N | *Chamaecyparis obtusa* | 61 | 552 | Iida city (Nagano) | 35.51N, 137.82E |
| 20 | N | *Chamaecyparis obtusa* | 76 | 600 | Iida city (Nagano) | 35.51N, 137.82E |
| 21 | A | *Chamaecyparis obtusa* | 77 | 210 | Nagoya city(Aichi) | 35.18N, 136.91E |
| 22 | A | *Chamaecyparis obtusa* | 118 | 277 | Nagoya city(Aichi) | 35.18N, 136.91E |
| 23 | N | *Chamaecyparis obtusa* | 250 | 511 | Iida city (Nagano) | 35.51N, 137.82E |
| 24 | N | *Chamaecyparis obtusa* | 332 | 640 | Miyata village (Nagano) | 35.77N, 137.95E |
| 25 | N | *Chamaecyparis obtusa* | 450 | 621 | Iida city (Nagano) | 35.51N, 137.82E |
| 26 | N | *Chamaecyparis obtusa* | 496 | 820 | Iida city (Nagano) | 35.51N, 137.82E |
| 27 | N | *Chamaecyparis obtusa* | 539 | 774 | Nirasaki city (Yamanashi) | 35.71N, 138.45E |
| 28 | O | *Chamaecyparis obtusa* | 584 | 792 | Ikaruga town (Nara) | 34.61N, 135.73E |
| 29 | O | *Chamaecyparis obtusa* | 683 | 832 | Okuwa village (Nagano) | 35.68N, 137.66E |
| 30 | O | *Chamaecyparis obtusa* | 709 | 890 | Okuwa village (Nagano) | 35.68N, 137.66E |
| 31 | N | *Chamaecyparis obtusa* | 717 | 919 | Iida city (Nagano) | 35.51N, 137.82E |
| 32 | N | *Chamaecyparis obtusa* | 719 | 1138 | Iida city (Nagano) | 35.51N, 137.82E |
| 33 | O | *Chamaecyparis obtusa* | 802 | 1159 | Okuwa village (Nagano) | 35.68N, 137.66E |
| 34 | A | *Sciadopitys verticillata* | 896 | 1116 | Inazawa city (Aichi) | 35.25N, 136.78E |
| 35 | A | *Chamaecyparis obtusa* | 908 | 1196 | Inazawa city (Aichi) | 35.25N, 136.78E |
| 36 | A | *Sciadopitys verticillata* | 1051 | 1169 | Inazawa city (Aichi) | 35.25N, 136.78E |
| 37 | A | *Sciadopitys verticillata* | 1068 | 1207 | Inazawa city (Aichi) | 35.25N, 136.78E |
| 38 | A | *Chamaecyparis pisifera* | 1087 | 1386 | Inazawa city (Aichi) | 35.25N, 136.78E |
| 39 | A | *Sciadopitys verticillata* | 1100 | 1331 | Inazawa city (Aichi) | 35.25N, 136.78E |
| 40 | L | *Chamaecyparis obtusa* | 1120 | 1930 | Nakatsugawa city (Gifu) | 35.49N, 137.50E |
| 41 | A | *Chamaecyparis obtusa* | 1126 | 1224 | Inazawa city (Aichi) | 35.25N, 136.78E |
| 42 | A | *Sciadopitys verticillata* | 1134 | 1267 | Inazawa city (Aichi) | 35.25N, 136.78E |





| 43 | L | *Chamaecyparis obtusa* | 1139 | 1978 | Okuwa village (Nagano) | 35.68N, 137.66E |
|---|---|---|---|---|---|---|
| 44 | A | *Sciadopitys verticillata* | 1148 | 1221 | Inazawa city (Aichi) | 35.25N, 136.78E |
| 45 | A | *Chamaecyparis pisifera* | 1151 | 1417 | Inazawa city (Aichi) | 35.25N, 136.78E |
| 46 | N | *Chamaecyparis obtusa* | 1169 | 1656 | Nakatsugawa city (Gifu) | 35.49N, 137.50E |
| 47 | A | *Sciadopitys verticillata* | 1189 | 1351 | Inazawa city (Aichi) | 35.25N, 136.78E |
| 48 | A | *Sciadopitys verticillata* | 1193 | 1326 | Inazawa city (Aichi) | 35.25N, 136.78E |
| 49 | L | *Chamaecyparis obtusa* | 1197 | 1968 | Otaki village (Nagano) | 35.81N, 137.55E |
| 50 | A | *Sciadopitys verticillata* | 1213 | 1313 | Inazawa city (Aichi) | 35.25N, 136.78E |
| 51 | A | *Chamaecyparis obtusa* | 1218 | 1294 | Kiyosu city (Aichi) | 35.20N, 136.85E |
| 52 | A | *Cryptomeria japonica* | 1219 | 1380 | Inazawa city (Aichi) | 35.25N, 136.78E |
| 53 | A | *Chamaecyparis obtusa* | 1231 | 1374 | Inazawa city (Aichi) | 35.25N, 136.78E |
| 54 | A | *Chamaecyparis obtusa* | 1265 | 1419 | Kiyosu city (Aichi) | 35.20N, 136.85E |
| 55 | A | *Chamaecyparis obtusa* | 1276 | 1465 | Kiyosu city (Aichi) | 35.20N, 136.85E |
| 56 | A | *Chamaecyparis obtusa* | 1278 | 1384 | Kiyosu city (Aichi) | 35.20N, 136.85E |
| 57 | A | *Chamaecyparis obtusa* | 1290 | 1454 | Kiyosu city (Aichi) | 35.20N, 136.85E |
| 58 | A | *Chamaecyparis obtusa* | 1331 | 1510 | Kiyosu city (Aichi) | 35.20N, 136.85E |
| 59 | A | *Chamaecyparis obtusa* | 1348 | 1502 | Kiyosu city (Aichi) | 35.20N, 136.85E |
| 60 | A | *Chamaecyparis obtusa* | 1485 | 1611 | Kiyosu city (Aichi) | 35.20N, 136.85E |
| 61 | L | *Chamaecyparis obtusa* | 1689 | 1988 | Otaki village (Nagano) | 35.78N, 137.69E |
| 62 | L | *Cryptomeria japonica* | 1718 | 1993 | Otsu city (Shiga) | 35.02N, 135.85E |
| 63 | L | *Chamaecyparis obtusa* | 1723 | 1993 | Otsu city (Shiga) | 35.02N, 135.85E |
| 64 | L | *Chamaecyparis obtusa* | 1730 | 2005 | Agematsu town (Nagano) | 35.78N, 137.69E |
| 65 | L | *Chamaecyparis obtusa* | 1761 | 2005 | Agematsu town (Nagano) | 35.78N, 137.69E |
| 66 | L | *Chamaecyparis obtusa* | 1827 | 2005 | Agematsu town (Nagano) | 35.78N, 137.69E |
| 67 | L | *Chamaecyparis obtusa* | 1839 | 2005 | Agematsu town (Nagano) | 35.78N, 137.69E |


 *1. "A" = archeologically excavated wood; "N" = naturally buried logs; "O" = old architectural wood; "L" = living trees. *2. Minus and plus numbers indicate years BCE and CE, respectively. *3. Sample locations are shown by the latitude and longitude of the governmental offices in the municipalities where samples were collected, because most samples were not living trees for which the actual growth location can be identified.








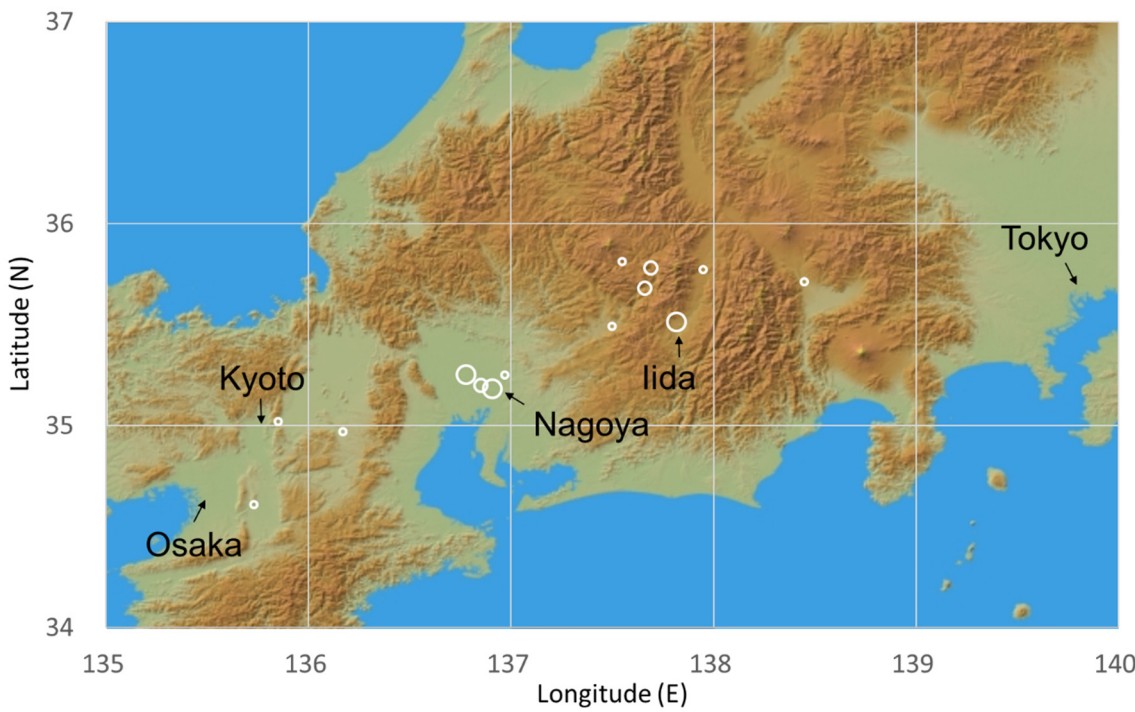

**Figure 1.** Sample sites of this study. Open circles indicate locations of wood samples, shown by the latitude and longitude of governmental offices in the municipalities where samples were collected, because most samples were not living trees for which the actual growth location can be identified. The size of the open circles indicates the sample numbers for each site (large = 10–15; medium = 4–9; small = 1–3). This figure was constructed using the JAXA AW3D30 DSM data map.





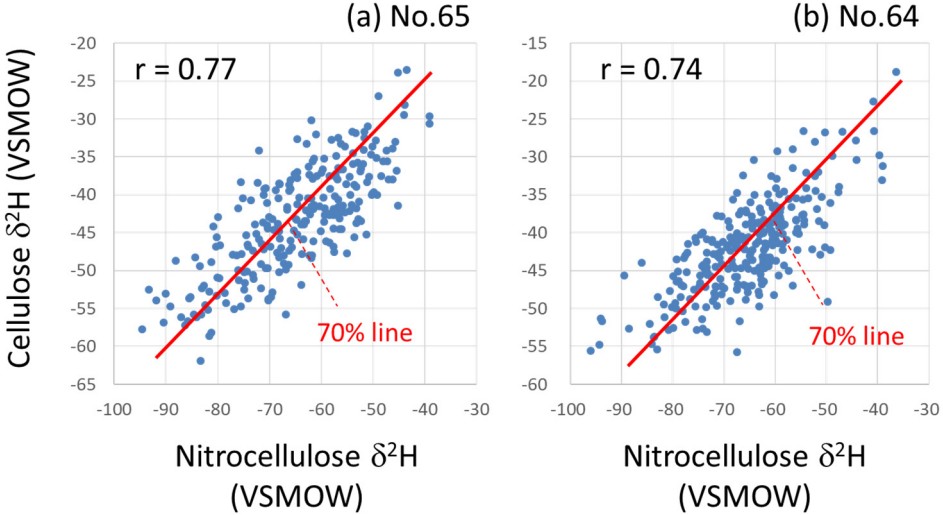


**Figure 2**. Correlations between hydrogen isotope ratios ($\delta^2$H) of tree-ring cellulose and nitrocellulose in two Japanese cypress trees. (a) Sample No. 65 and (b) Sample No. 64 in Table 1.





**Figure 3.** The tree-ring cellulose $\delta^{18}O$ and $\delta^2H$ time-series and their statistical properties. (a) Tree-ring cellulose $\delta^{18}O$ time-series of the 67 analyzed trees. (b) R-bar (average correlation coefficients between different individual trees), EPS (see text), and sample depth (tree sample numbers) in the corresponding periods for the tree-ring cellulose $\delta^{18}O$ dataset. (c) Tree-ring cellulose $\delta^2H$ time-series of the 67 analyzed trees. (d) R-bar, EPS, and sample depth for the tree-ring cellulose $\delta^2H$ dataset. All data in (b) and (d) were calculated for a total period of 51 yr, including 25 yr before and after the year shown.








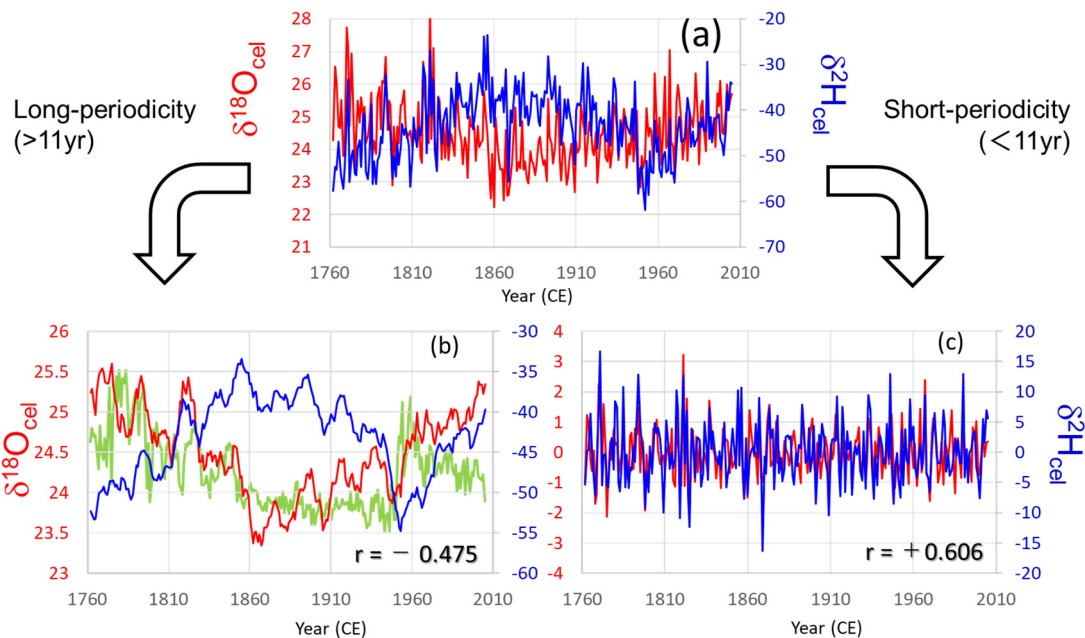

**Figure 4.** Comparison between tree-ring cellulose $\delta^{18}O$ (red) and $\delta^{2}H$ (blue) time-series in tree sample No. 65. (a) Raw data, (b) long-periodicity components (>11 yr; 11-yr running mean) with variations in tree-ring width (light green), and (c) short-periodicity components

(<11 yr; deviation from the 11-yr running mean).





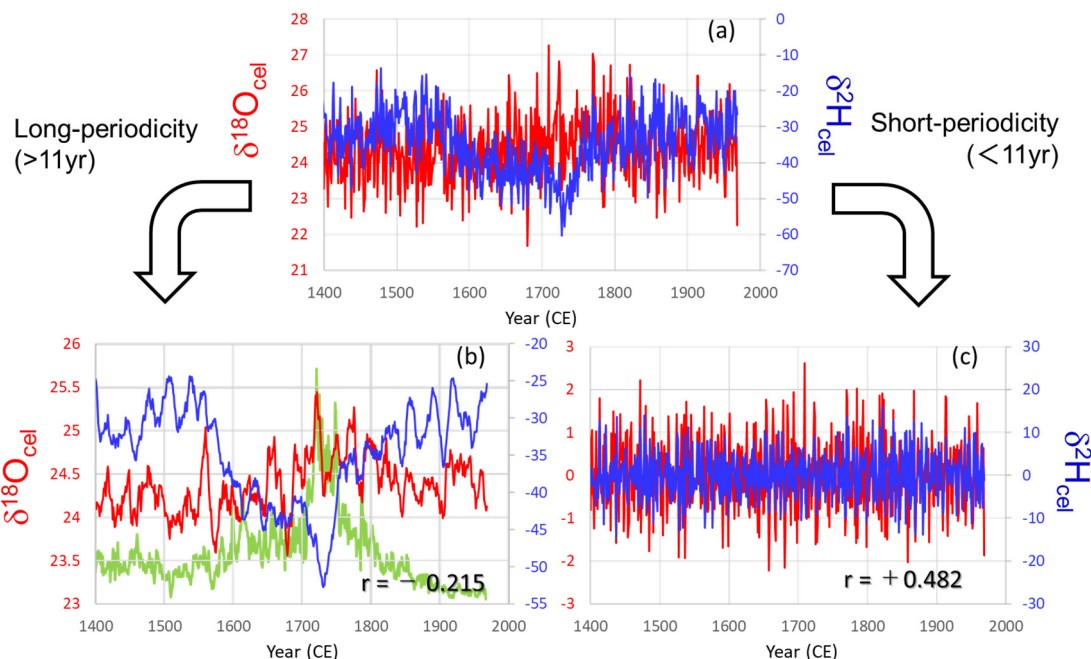

**Figure 5.** Comparison between tree-ring cellulose $\delta^{18}O$ (red) and $\delta^2H$ (blue) time-series in tree sample No. 49. (a) Raw data,
(b) long-periodicity components (>11 yr; 11-yr running mean) with the variations in tree-ring width (light green), and (c) short-periodicity components (<11 yr; deviation from the 11-yr running mean).





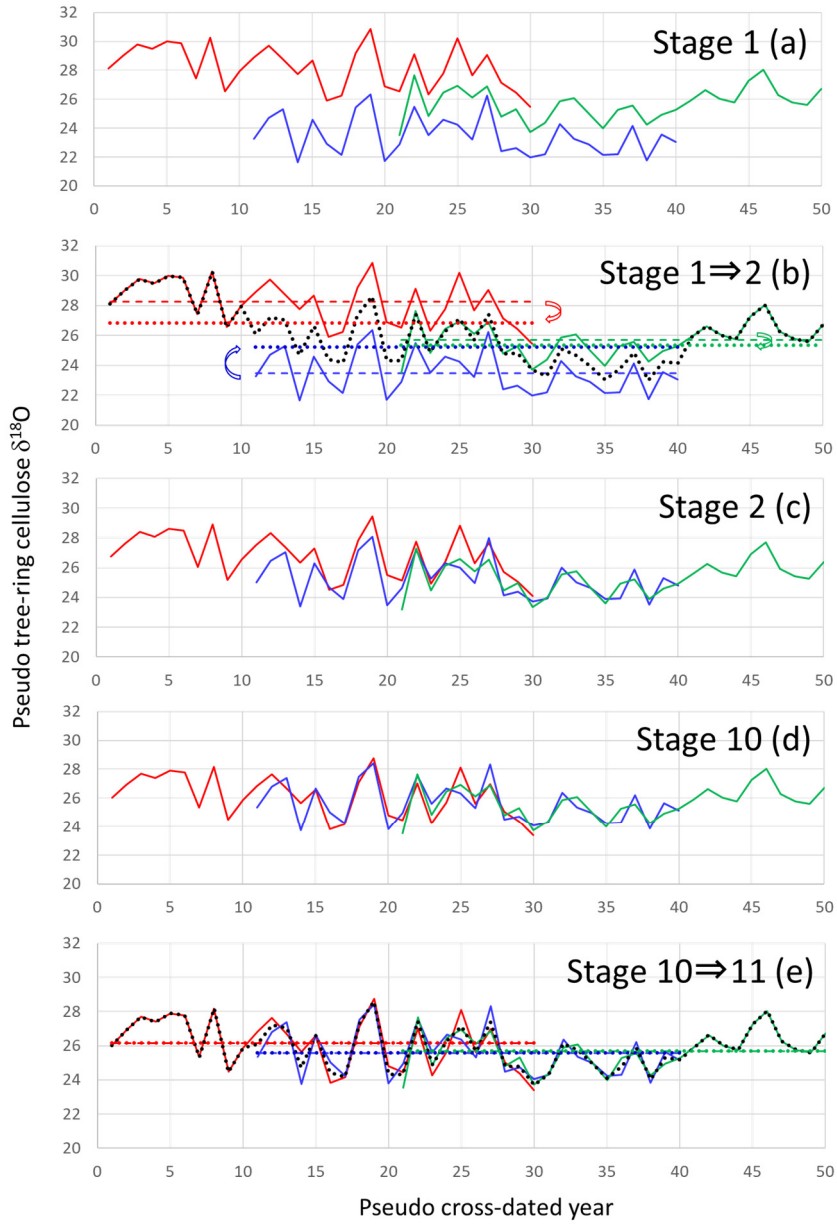

**Figure 6.** The iterative calculation method used to smoothly combine many individual tree-ring time-series with large level offsets. (a) Pseudo-cross dated tree-ring cellulose δ¹⁸O time-series (Stage 1). (b) Averaging of all time-series in Stage 1 (black dotted line) and offsetting of individual time-series in Stage 1 to the position where the average of the individual time-series (colored dashed lines) becomes equal to the average of the black dotted line during the corresponding period (colored dotted lines), by retaining the (c) original temporal patterns (Stage 2). (d) Result of the iterative calculation from Stage 10 and the (e) procedure from Stages 10 to 11. We obtain a unique combined time-series when the dotted lines become equal to the dashed lines, within a finite number of iterations.







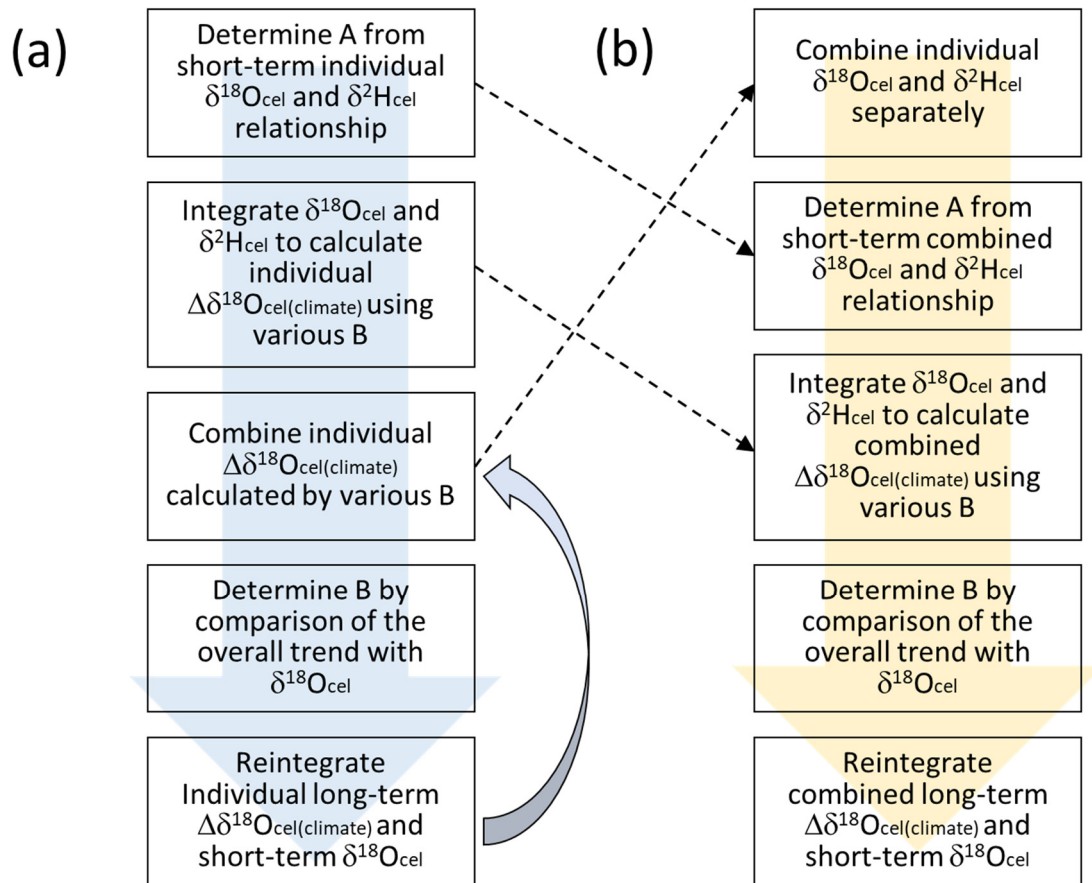

**Figure 7.** The two procedures used to establish the smoothly combined climatological component of the variations in the tree-ring cellulose

$\delta^{18}O$ data ($\Delta\delta^{18}O_{cel(climate)}$). Although the order of procedures is different in (a) and (b), the results are the same if we assume unique A and

B values for all individual trees.





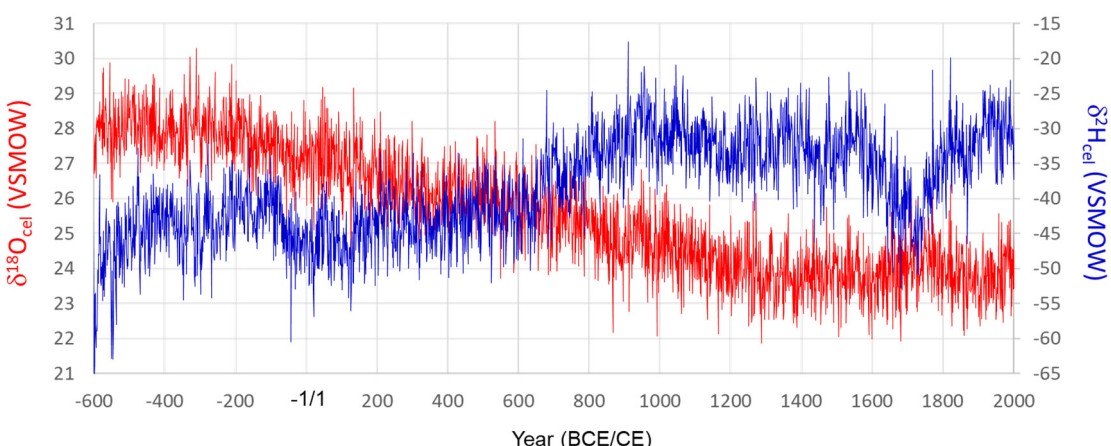

**Figure 8.** Smoothly combined individual time-series of $\delta^{18}O_{cel}$ and $\delta^2H_{cel}$ from Fig. 3a and c, using the iterative calculation method shown in Fig. 6. The combined time-series is after 1000 iterations (Stage 1000).



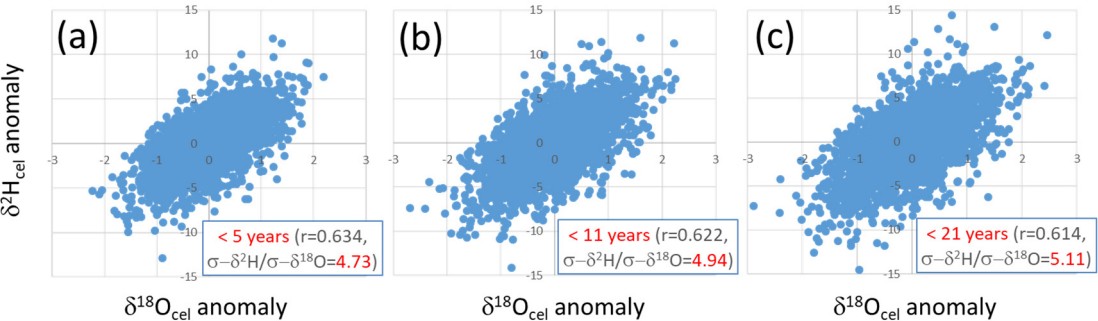

**Figure 9.** Relationships between short-periodicity components of $\delta^{18}O_{cel}$ and $\delta^2H_{cel}$ that are (a) <5 yr, (b) <11 yr, and (c) <21 yr, defined as the anomalies from the 5-, 11-, and 21-year running means of the combined $\delta^{18}O_{cel}$ and $\delta^2H_{cel}$ time-series in Fig. 8, respectively.





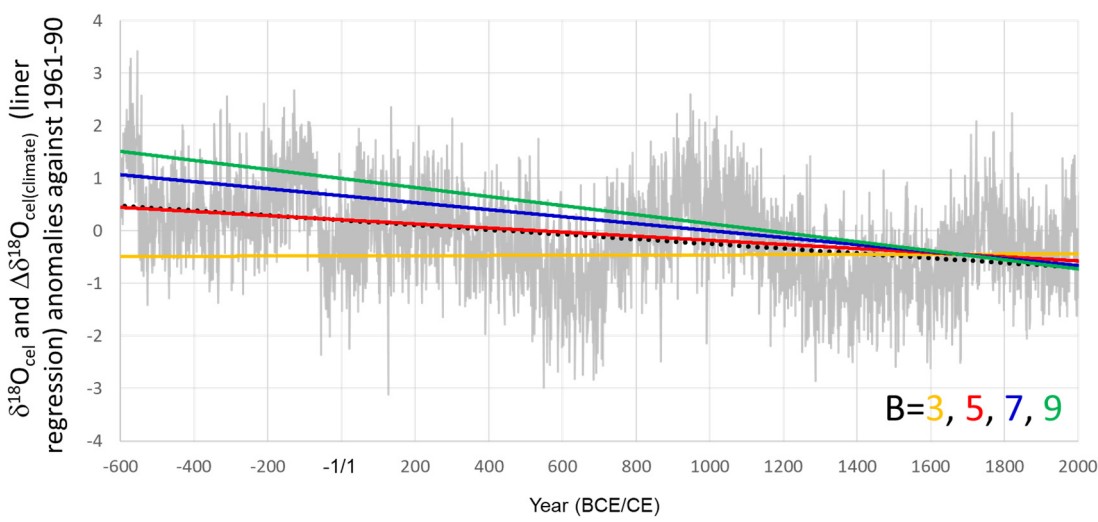


**Figure 10.** Linear regressions of $\Delta\delta^{18}O_{cel(climate)}$ calculated using 3 (orange line), 5 (red line), 7 (blue line), and 9 (green line) as B values, with A = 5 in Eq. (21), and the simple averages of all tree-ring cellulose $\delta^{18}O$ data in Fig. 3a (gray line) with a linear regression (black dotted line).

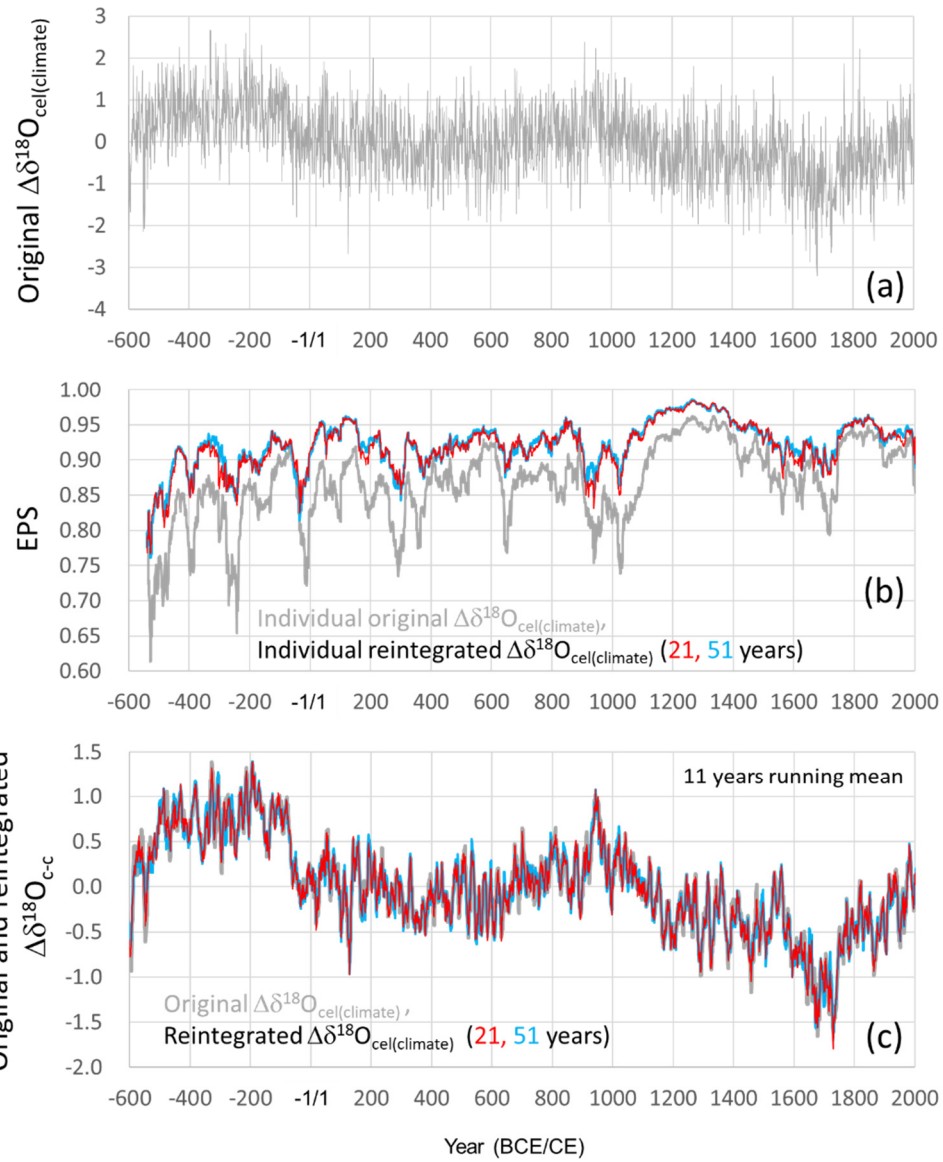


**Figure 11.** Calculation of the climatological component of the tree-ring cellulose $\delta^{18}O$ data ($\Delta\delta^{18}O_{cel(climate)}$). (a) Time-series of $\Delta\delta^{18}O_{cel(climate)}$ calculated using Eq. (21) with A = 5.0 and B = 5.4, and the smoothly combined time-series of $\delta^{18}O_{cel}$ and $\delta^2H_{cel}$ in Fig. 8. (b) EPS values for the individually calculated time-series of $\Delta\delta^{18}O_{cel(climate)}$ (gray line) and the individually calculated and reintegrated time-series between the long-periodicity component of $\Delta\delta^{18}O_{cel(climate)}$ and short-periodicity component of $\delta^{18}O_{cel}$, with 21 yr (red line) and 51 yr (blue line) as the

thresholds between long- and short-periodicity. (c) 11-yr running mean of the time-series of $\Delta\delta^{18}O_{cel(climate)}$ in Fig. 11a (gray line) and those of the reintegrated time-series between the long-periodicity component in $\Delta\delta^{18}O_{cel(climate)}$ in Fig. 11a and short-periodicity component in the combined $\delta^{18}O_{cel}$ in Fig. 8, with 21 yr (red line) and 51 yr (blue line) as the threshold between long- and short-periodicity. EPS values in (b) were calculated for a total period of 51 yr, including 25 yr before and after the year shown.





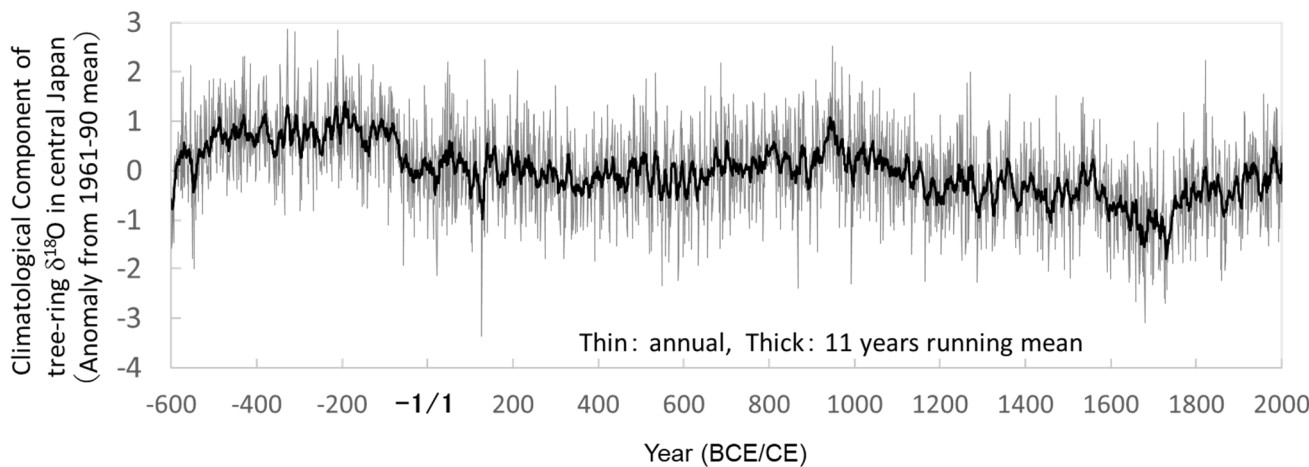


**Figure 12.** The final combined time-series of the climatological component in variations of tree-ring cellulose $\delta^{18}O$ data ($\Delta\delta^{18}O_{cel(climate)}$). This is the reintegrated time-series between the long-periodicity component in the original $\Delta\delta^{18}O_{cel(climate)}$ data in Fig. 11a and the short-periodicity component in the combined $\delta^{18}O_{cel}$ data in Fig. 8, with 21 yr as the threshold between long- and short-periodicity.






**Figure 13.** (a–c) Average values of the monthly mean temperature (red), precipitation (blue), and mean relative humidity (green) from 1901–2005 CE. (d–f) Correlation coefficients of variations between $\Delta\delta^{18}O_{cel(climate)}$ in Fig. 12 and the monthly mean temperature (red), precipitation (blue), and mean relative humidity (green) from 1901–2005 CE. The meteorological data were observed at Iida (a, d), Nagoya (b, e), and Kyoto (c, f) (see Figs 1 and 14b), and recorded in the database of the Japan Meteorological Agency. Asterisks in d, e, and f indicate statistical significance at the >99% level (p < 0.01).





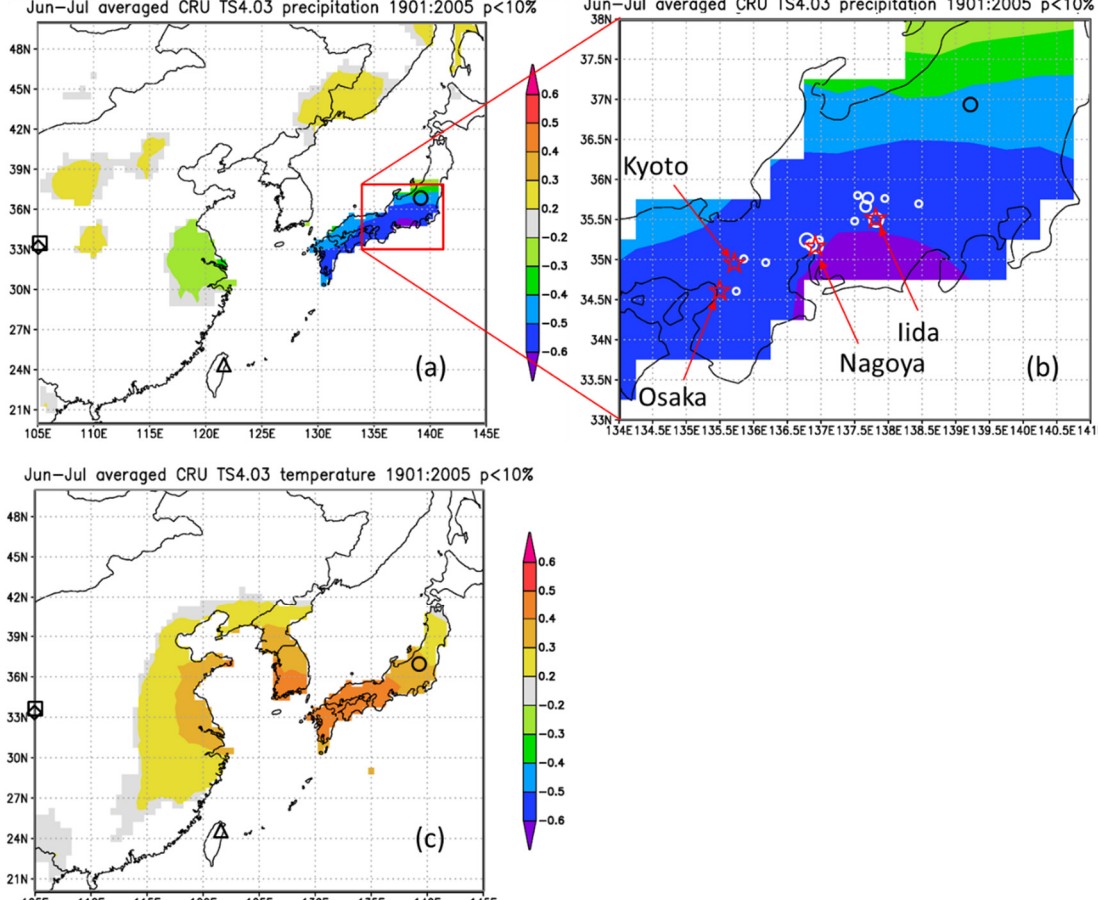

**Figure 14.** Maps of the spatial correlation (p < 10%) of the $\Delta\delta^{18}O_{cel(climate)}$ in Fig. 12 with (a–b) June–July precipitation and (c) June–July mean temperature from 1901–2005 from CRU TS4.03, with sites of reference studies shown as an open square (Huangye Cave), diamond (Wanxiang Cave), triangle (Tsuifong Lake), and circle (Ozegahara peatland). For this analysis, we used Climate Explorer (http://www.knmi.nl/) of the Royal Netherlands Meteorological Institute (KNMI) (van Oldenborgh and Burgers, 2005). Sample locations and numbers (Table 1) are shown as white circles (large = 10–15; medium = 4–9; small = 1–3) in (b).







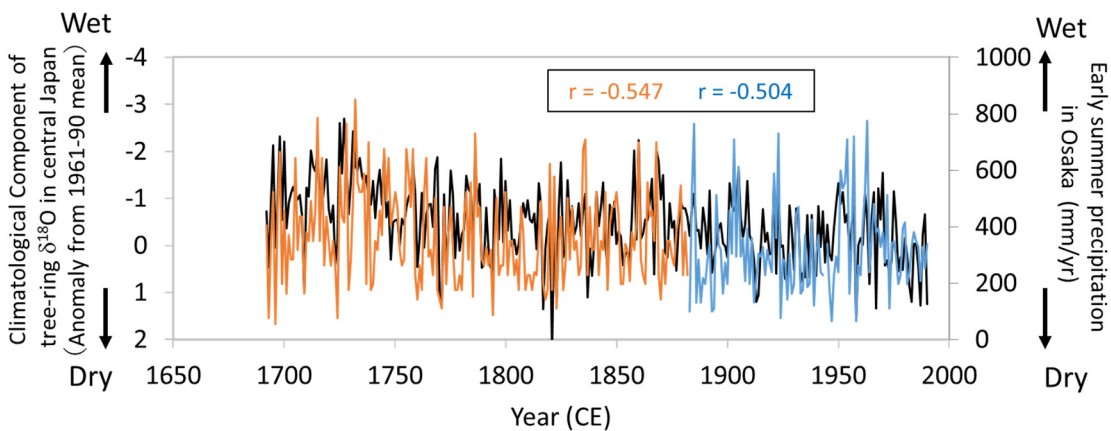

**Figure 15.** Comparisons of the climatological component of the tree-ring cellulose $\delta^{18}O$ ($\Delta\delta^{18}O_{cel(climate)}$) record in Fig. 12 (black line) with
diary-based annual reconstructions (orange line) and meteorological observations (blue line) of early summer precipitation at Osaka
(Mizukoshi, 1993). Note that the y-axis of $\Delta\delta^{18}O_{cel(climate)}$ is inverted to indicate higher precipitation toward the top of the figure.







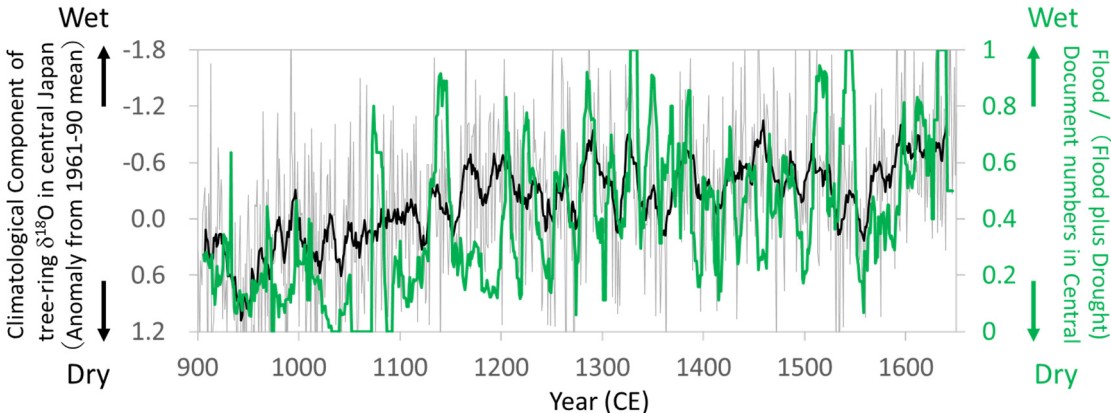

**Figure 16.** Comparisons of the climatological component of the tree-ring cellulose $\delta^{18}$O ($\Delta\delta^{18}$O$_{cel(climate)}$) record in Fig. 12 (gray line = annual; black line = 11-yr running mean) with the proportion of flood-related documents (11-yr running mean) of total flood- and drought-

related documents (11-yr running mean) during the summer season (JJA) in central Japan (mostly in and around Kyoto) (Fujiki, 2007). Note that the y-axis of $\Delta\delta^{18}$O$_{cel(climate)}$ is inverted to indicate higher precipitation toward the top of the figure.







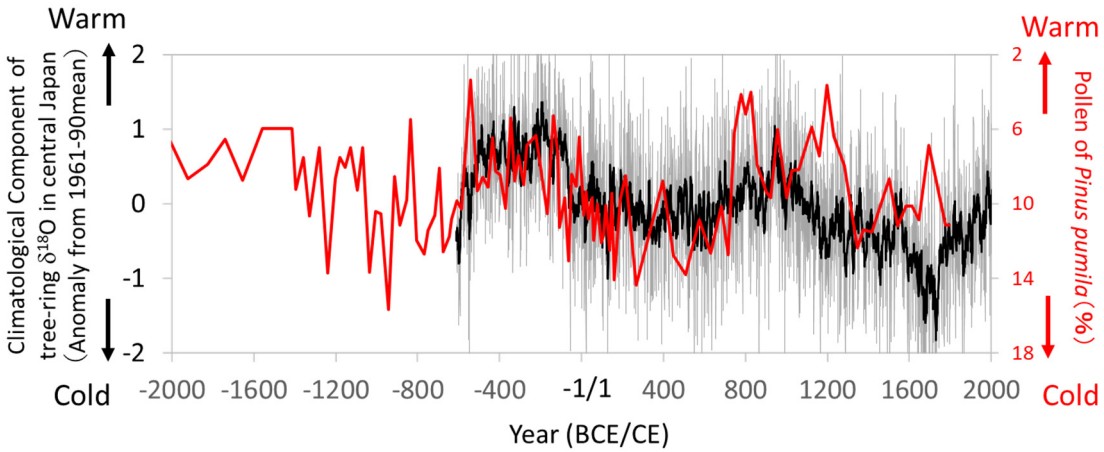

**Figure 17.** Comparisons of the climatological component of the tree-ring cellulose $\delta^{18}O$ ($\Delta\delta^{18}O_{cel(climate)}$) record in Fig. 12 (gray line = annual; black line = 11-yr running mean) with the pollen percentage of a cold region pine (*Pinus pumila*) in the Ozegahara peatland

(Sakaguchi, 1983, 1989). Location of the pollen record is shown in Fig.14. Given that *P. pumila* becomes dominant during cold periods, the y-axis for the pollen percentage is inverted to indicate warmer conditions toward the top of the figure.




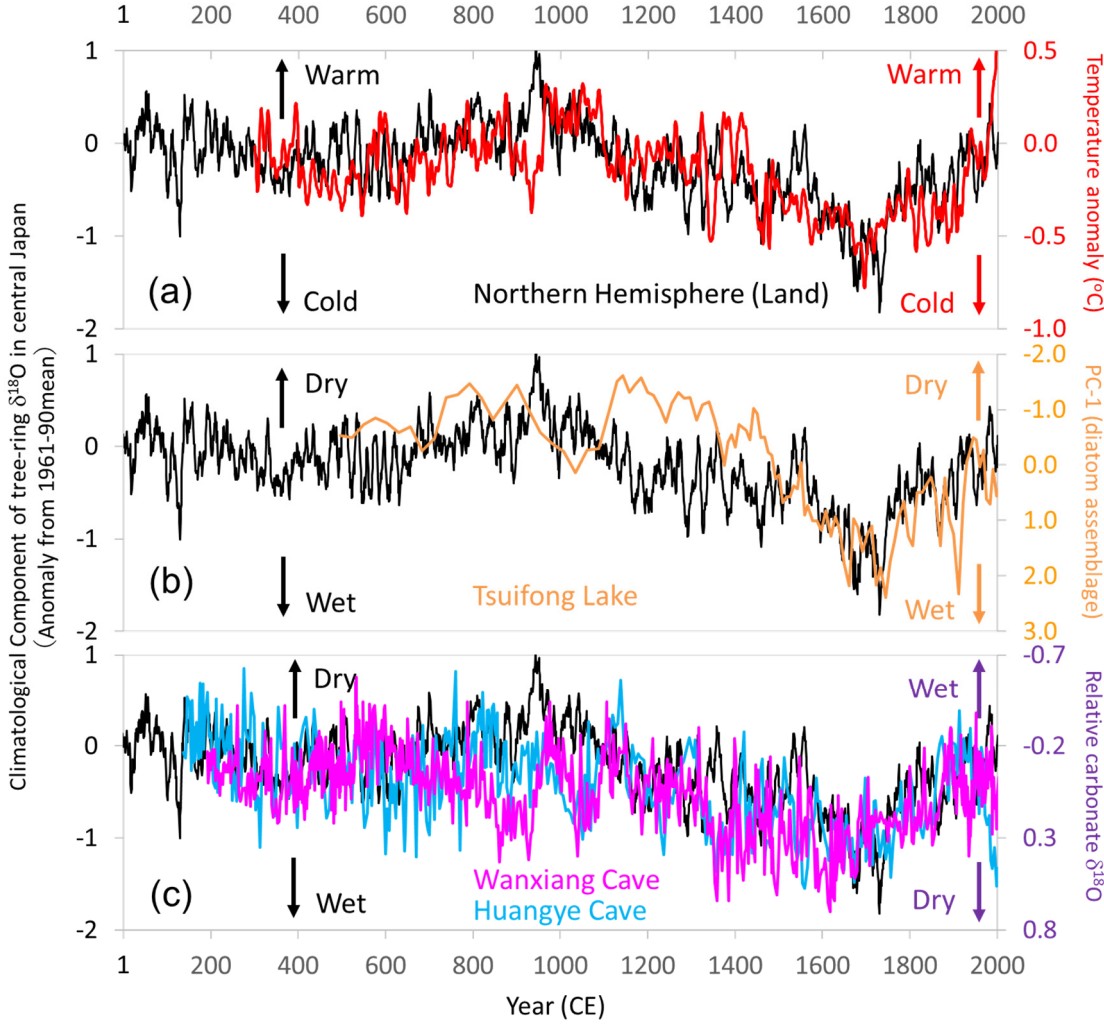

**Figure 18.** Comparison of centennial- to millennial-scale variations of the climatological component of tree-ring cellulose $\delta^{18}O$ ($\Delta\delta^{18}O_{cel(climate)}$) in central Japan with global and East Asian climate reconstructions. The time-series of 11-yr running means for Fig. 12 (black line) with (a) Air temperature over the Northern Hemisphere defined as NH EIV-land (Mann et al., 2008) (red line), (b) Precipitation inferred from principal component analysis (PC1) of diatom assemblages from Tsuifong Lake in southeastern China (Taiwan) (Wang et al., 2013) (orange line), and (c) Precipitation inferred from carbonate $\delta^{18}O$ values of speleothems from Wanxiang (pink line) and Huangye (blue line) caves in northwestern China (Zhang et al., 2008; Tan et al., 2010). Note that the y-axes are reversed between the speleothem data and $\Delta\delta^{18}O_{cel(climate)}$ in (c). Locations of the records in (b) and (c) are shown in Fig. 14.