# Peer review of "A 2600-year summer climate reconstruction in central Japan by integrating tree-ring stable oxygen and hydrogen isotopes"

_Climate of the Past, 2020_

## Referee Comment (RC1) · Anonymous Referee #1 · 6 Mar 2020

A nice paper. This manuscript presents two very long tree-ring stable isotopes (oxygen and hydrogen) series by measuring 67 different kind of wood samples, including living tree, archaeological wood and buried logs, over central Japan, a pretty remote region from which such information is novel. An important innovation of this manuscript is the authors created a novel method to remove age trend in tree-ring d18O by integrating physiological mechanism and correlation between the d18O and d2H. The manuscript, which I believe to have already reviewed by top journals, and is almost mature to be published. Still, however, some points should be solved, and a general revision of

the reorganization (see lines 69-84 as example), although not bad in its current state, would certainly strengthen the value of the paper.

Special comments are as follow. ïĄň Title could be simplified, which may make general readers impressive. For example, A 2600-year summer climate reconstruction in central Japan by integrating tree-ring stable oxygen and hydrogen isotopes. ïĄň Lines 25-26, change "living old trees, excavated archeological wood, old architectural wood, and naturally buried logs" to "living trees, archeological wood and buried logs" ïĄň Lines 46-49, to my knowledge, any method cannot reserve low-frequency climatic signals of tree-ring width/density fully. ïĄň Line 65 delete "summer" ïĄň Lines 67-68, cite studies with long tree-ring d18O chronology in Asia, such as Liu et al., 2017 ïĄň Lines 69-84 describe methods on cellulose extraction and removing age trends. It's better to move them in Section 2. ïĄň Line 94, hundreds of rings

---

## Referee Comment (RC2) · Anonymous Referee #2 · 14 Apr 2020

SYNOPTIC COMMENT The authors use of d2H and d18O tree-ring series for reconstruction of central Japan hydroclimate. Combining composite data set constitutes a serious technical challenge. The authors selected stem segments mostly of Japanese cypress from living trees, excavated archeological wood, architectural wood and naturally buried logs.

They propose an iterative calculation method to merge 67 series from the various types of wood samples, including the buried archeological and construction wood pieces, and a tentatively quantitative method (factors A and B) to calculate past climate based on

d2H and d18O values, using a suite of equations derived from Roden's et al (2000). However, they did not nitrate their samples prior to analysing the d2H values of tree-ring cellulose, so that the exchangeable H is included in their analyses. The simple determination of d2H values on cellulose can generate artefacts. Additionally, the number of trees studied for d2H results are significantly lower than for the d18O determination, and the expressed population signals obtained for the composite d2H series are too low (Fig. 3, b, d). The fact that the d2H and d18O series do not derive from the same populations of trees may generate artefacts. Another point is that the authors did not evaluate the reliability of the isotopic signals for the buried pieces of wood, but alteration of cellulose can occur due to microbial activities during long periods of burial.

Overall, the article is lengthy for what it brings, but generally clearly written. The discussion of the low-frequency trends (long-periodicity variations) is confusing. The authors interpret them unguardedly as age trends, without presenting supporting arguments, and then they bring up the option of these trends possibly relating to changes in growth rates (lines 149-150; 160-165). This potential interpretation implies that environmental conditions may have generated these trends, at least partly. Moreover, the use of ring width for specifically deducing the cause of inverse d2H and d18O trends is risky because in many cases, the isotopic and ring width series do not respond to the same environmental factors.

Another important point is that some of the sampled populations of trees belong to forests exposed to human perturbations; such sites are not suitable for producing isotopic series to be used for climatic reconstruction.

Concluding that (1) d2H analyses would be more reliable if performed on nitrated cellulose, and (2) memory effects occur when performing online pyrolysis, are not new findings and do not bring constructive information in this field of research. Furthermore, as mentioned above, it underlines the fact that 50% of the data used for evaluating paleoclimate is faulty, and weakens the basis for the final reconstruction.

Overall, given the purpose of the article and the unfortunate non-rigorous sample selection and treatment, CP should not accept this article.

SPECIFIC POINTS Line 30 – replace However by In addition.

Line 75 – Please provide a minimum of details about the direct cellulose extraction of 1-mm thick wood samples so the reader does not have to read two other articles to find out. Do you mean that all stem segments were dissected into suite 1 mm-thick samples regardless of ring width and age?

Line 76 – Please explain what are 'level offsets'.

Line 80 – Simultaneous (?) measurements of d2H and d18O values? How possible with good precision? Using more than one standards is required for a good calibration (two end members with distant isotopic values defining a range broader thant the measured isotopic ranges, and a third standard as an intermediate checkpoint), but the described analytical procedure does not mention this required approach.

Lines 86-87 – Please modify text. . . for reconstructing climate over the past 2,600 yr. . .

Lines 91-96 what are the average, minimum and maximum ring widths of the studied samples; this information will help follow the wood slicing procedure of next section.

Lines 96-97 – Please briefly explain how the new tree-ring d18O time series were used for dating rings. Usage of a statistically strong constructed and multiply verified d18O suite as dating method? How widely is this applicable? For which geographical area was the dating series constructed? What is the operating time resolution on which the comparison is used?

Lines 102-107 – Scientists have recognized for a long time that the production of tree-ring d2H series to be coherent requires nitrating cellulose, so that only the C-bond hydrogen is analysed (Epstein et al., 1976; a reference they use and list). Otherwise, exchangeable H may blur true environmental effects. It seems here that the authors have chosen to save time by analysing simultaneously d2H and d18O in non-nitrated

cellulose. No surprise that they conclude they need to improve their analytical procedure (lines 485-487). However, this information has been available to scientists since 1976, with a methods proposed for improving the throughput and reducing the amount of material required back in 2006 and 2009 (Filot et al., 2006; Sauer, 2009; rapid comm. mass spectrom.).

Line 105 – Strict rigor would require indicating the true significant numbers for precisions (reproducibility), i.e., 0.1 or 0.2‰ and 1 or 2‰ for d18O and d2H values, respectively. In the light of the moderate correspondence between nitrated cellulose and cellulose, and of the analytical protocol (only one standard, memory effects not dealt with, peak jumping), it seems hardly conceivable that the d2H precision and accuracy would be of 1‰ it is likely no better than 3‰Ėven with limited effects from OH-exchangeable fraction, the analytical precisions are rarely better than 2‰ (Filot et al., 2006; Sauer, 2009).

Equations 1 and 2 – The ‰ sign should be on the left of the equations, near the delta notation. Otherwise, ‰ x 1000 implies no change in the reported values.

Lines 112-113 - Why all the cellulose samples could not be nitrated? Not enough material extracted from wood? The authors decided to follow an alternative approach, not clearly defined (temperature, time of equilibration), but apparently different than the Filot approach, so that their cellulose and nitrated cellulose only show correlations (r) between 0.74 and 0.77, which is significantly lower than the correspondence obtained using the rigorous protocol of Filot et al (0.94). This compromise is not ideal when producing d2H series destined to climatic reconstruction.

Lines 118-119 – The authors should revisit this statement and write with more nuance, because they sacrifice on the reliability of the d2H series by analysing cellulose instead of nitrated cellulose, or by apparently using an alternative protocol that unfortunately does not perform as well as previous equilibration protocols documented in the literature (Filot et al., 2006; Sauer, 2009).

Lines 199-121 – The memory effects are well known when dealing with online pyrolysis systems, and there are several ways to avoid analytical artefacts due to them. Possibilities include placing a blank (empty capsule) between each samples in the carousels, or analyzing samples in triplicates, or a combination of the two approaches, etc. The appropriate analytical protocol with the instrument should be decided upfront, prior to producing the results. Unfortunately, again, the authors underline the issue after conducting all analyses, but truly this issue could have been easily dealt with prior to producing the isotopic series.

Lines 125-129 – This text and Figure 3 do not inform the reader about the distribution of the woo types. Which isotopic series derive from buried pieces of wood? Departures from real values are reported to occur for altered cellulose/wood (Yapp, 2001; Mancini et al., 2003; Savard et al., 2012).

Mancini, S.A., Ulrich, A.C., Lacrampe-Couloume, G., Sleep, B., Edwards, E.A., Lollar, B.S., 2003. Carbon and hydrogen isotopic fractionation during anaerobic biodegradation of benzene. Applied and Environmental Microbiology 69, 191–198.

Yapp, C., 2001. Rusty relics of earth history: iron(III) oxides, isotopes, and surficial environments. Annual Review of Earth and Planetary Sciences 29, 165–199.

Savard, M. M., Bégin, C., Marion, J., Arseneault, D., and Bégin, Y., 2012. Evaluating the integrity of C and O isotopes in sub-fossil wood from boreal lakes, Palaeogeogr. Palaeoclim. Palaeoecol., 348–349, 21– 31.

Lines 129 – What are the indications that these are age trends? Are trends visible on all individual tree segments prior to combining them? Or are they visible after combining them? In the later case, the authors should consider discussing the possibility of an artefact to the treatment of the data.

Lines 136-139 – The authors clearly state here why their d2H series are not reliable or suitable for climatic reconstruction. Ideally, they should not be used in the following

parts of the article (or in the article).

Section 3.2 – The entire discussion about the supposed 'age trends' is misleading. What are the arguments supporting this interpretation? If growth rates correlate with d2H and d18O, inversely and directly, what are the most logical environmental reasons for that? Why do d2H and d18O trends inversely correlate (assuming that the d2H trends reflect something real)? Any possible mechanisms in teleconnection that could explain coeval long-term changes in the three proxies (growth, d2H and d18O)?

Lines 214-215 – It seems that the sites selected for this research are not suited for climatic reconstruction.

Lines 251-252 & Eq 18 – The physiological effects does not always generate a negative relationship between d2H and d18O series. Is it not right?

Lines 258 & 267 – Using constant A and B values implies multiple big assumptions.

Line 296-298 – Another big assumption that this simple combination cancels out the inter-tree average offsets.

Section 3.7 – How can the authors attest that this approach does not generate artefacts at the point of junction between series (e.g., Gagen et al., 2012).

Gagen, M., McCarroll, D., Jalkanen, R., Loader, N. J., Robertson, I., and Young, G. H. F., 2012. A rapid method for the production of robust millennial length stable isotope tree ring series for climate reconstruction, Global Planet. Change, 82–83, 96-103.

Section 3.8 – It seems that there are several short cuts slid in the procedure for attempting to correct for limitations introduced by the analytical approach (lines 367-371). Since there is no true comparison with a fully rigorous approach, the assessment of the procedure is impossible. The comparisons made with reconstructions from other proxies show significant departures and do not allow assessing the proposed procedure (section 3.11).

Line 359 – Note clear... as... Please rewrite.

Lines 394-396 – The idea is with paying attention, but unfortunately, the basic sampling and analytical procedures selected for this research are not rigorous enough to allow evaluating the approach in this article.

Table 1 and Figure 2 – It seems that the term 'sample' here refers to stem segments.

Figure 2 – 70% line? Not clear what it is and what it means?

The number of figures is high; perhaps some of them would find a better place in a supplement of information, for examples figures 9, 13, 14.

---

## Author Comment (AC1) · 12 May 2020

We will present our responses to the referee#1's comment (RC1) one by one in the following order. (1) comments from Referees, (2) author's response, (3) author's changes in manuscript.

(1) A nice paper. This manuscript presents two very long tree-ring stable isotopes (oxygen and hydrogen) series by measuring 67 different kind of wood samples, including living tree, archaeological wood and buried logs, over central Japan, a pretty remote re-

gion from which such information is novel. An important innovation of this manuscript is the authors created a novel method to remove age trend in tree-ring d18O by integrating physiological mechanism and correlation between the d18O and d2H. The manuscript, which I believe to have already reviewed by top journals, and is almost mature to be published. Still, however, some points should be solved, and a general revision of the reorganization (see lines 69-84 as example), although not bad in its current state, would certainly strengthen the value of the paper. (2) Thank you very much for your deep understanding and high evaluation for our paper. (3) We will revise our paper according to your comments below.

Special comments are as follow.

(1) Title could be simplified, which may make general readers impressive. For example, A 2600-year summer climate reconstruction in central Japan by integrating tree-ring stable oxygen and hydrogen isotopes. (2) Thank you for your comment. (3) We will revise the title as you recommended.

(1) Lines 25-26, change "living old trees, excavated archeological wood, old architectural wood, and naturally buried logs" to "living trees, archeological wood and buried logs" (2) Thank you for your comment. (3) According to your comment, we will simplify the description of wood types.

(1) Lines 46-49, to my knowledge, any method cannot reserve low-frequency climatic signals of tree-ring width/density fully. (2) Thank you for your suggestion. (3) We will modify the sentence according to your suggestion.

(1) Line 65 delete "summer" (2) Thank you for your comment. (3) We will delete "summer".

(1) Lines 67-68, cite studies with long tree-ring d18O chronology in Asia, such as Liu et al., 2017 (2) Thank you for your comment. (3) We will cite the paper of Liu et al. (2017) here.

(1) Lines 69-84 describe methods on cellulose extraction and removing age trends. It's better to move them in Section 2. (2) Thank you for your suggestion. (3) We will move most part of them to Section 2 and leave only essential idea here.

(1) Line 94, hundreds of rings (2) Thank you for your comment. (3) We will modify the word according to your suggestion.

That is all. Thank you for your comments.

---

## Author Response (AR1)

07 Aug 2020

Dear Editor,

I am sending herewith two PDF files of our revised manuscript entitled as "A 2600-year summer climate reconstruction in central Japan by integrating tree-ring stable oxygen and hydrogen isotopes", which has been slightly changed from original one according to the suggestion by Reviewer#1, and supplementary materials for the possible publication in CP.

I am very sorry to delay sending the revision until now due to the COVID19 problems and my personal unhappiness.

In the PDF files, modified and/or newly added portions are shown with red-colored characters in the revised manuscript and supplementary materials. Our revised manuscript has not passed an English editing by some native speakers, because we heard that some experts in CP journal will check our English of manuscript carefully after acceptance of the manuscript.

Following pages of responses to Editor and two reviewers includes point-by-point responses to the all comments from reviewers and you.

We hope that the revisions of manuscript are satisfactory for you and reviewers.

Best regards,
Takeshi

Takeshi Nakatsuka, PhD. Professor
Graduate School of Environmental Studies, Nagoya University, Japan

Responses to Editor's comments.

We will present our responses to the Editor's comments in the following order. (1) *comments from Referees*, (2) author's response, (3) author's changes in manuscript.

(1) Thank you for the responses to the reviewers comments. As you note there is some disagreement between the two reviewers, where #2 suggests rejection. However, based on the novelty of the data from this region, I will give you the opportunity to revise the manuscript. In the revision process, please make sure to attend to the issues raised by ref #2, including the potential impact of disturbed growth environments of the trees since this may have an impact on the final reconstruction [A]. Personally, I would also like to see some more information of the dating of the individual species, such as how the TRW data was dated (against what data), and what method was used to date the isotopes [B]. In general, I think it is good not to hide the limitations of your study, as this information will be very helpful for further research. Good luck with the revision, I look forward to the next version.

(2) Thank you very much for your encouragement to revise our paper. And, I am very sorry to delay sending the revision until now due to the COVID19 problems and my personal unhappiness. As for your two big comments marked by underlines above, we have added new figures and information as below. We hope that the revisions are satisfactory for you. *Modified and/or newly added portions are shown with red-colored characters in the revised manuscript.

*Our revised manuscript has not passed an English editing by some native speakers, because we heard that some experts in CP journal will check our English of manuscript carefully after acceptance of the manuscript.

(3)   [A] The utilization of trees influenced by human disturbance is the key issue of this paper because trees excavated from archaeological remains, that is only one source of our study for the distant past in Japan, are usually influenced by human activities. In order to show that our method can really remove the physiological effects, caused by the human disturbance, from tree-ring cellulose d18O time-series, we added Appendix A to show how human disturbances influence the tree-ring cellulose d18O and d2H physiologically (Fig. A1 and A2) and demonstrate how our method successfully removes the physiological effects by comparing three cases of nearly millennial length of independent trees as examples (Fig. A3).

   [B] We added a supplementary figure (Fig. S1) to show that we have dated successfully all stem segments in this study by tree-ring cellulose d18O with the help of traditional TRW

based dating. Relating information on dendrochronological dating are written in the caption of Fig. S1.

Thank you for your consideration of publishing this manuscript in CP.

Responses to Referee #1's comments.

We will present our responses to the referee's comments one by one in the following order. (1) *comments from Referees*, (2) author's response, (3) author's changes in manuscript (lines in the revised manuscript).

(1) *A nice paper. This manuscript presents two very long tree-ring stable isotopes (oxygen and hydrogen) series by measuring 67 different kind of wood samples, including living tree, archaeological wood and buried logs, over central Japan, a pretty remote region from which such information is novel. An important innovation of this manuscript is the authors created a novel method to remove age trend in tree-ring d18O by integrating physiological mechanism and correlation between the d18O and d2H. The manuscript, which I believe to have already reviewed by top journals, and is almost mature to be published. Still, however, some points should be solved, and a general revision of the reorganization (see lines 69-84 as example), although not bad in its current state, would certainly strengthen the value of the paper.*

(2) Thank you very much for your deep understanding and high evaluation for our paper.

(3) We have revised our paper according to your comments below.

Special comments are as follow.

(1) *Title could be simplified, which may make general readers impressive. For example, A 2600-year summer climate reconstruction in central Japan by integrating tree-ring stable oxygen and hydrogen isotopes.*

(2) Thank you for your comment.

(3) We have revised the title as you recommended (lines 1-2).

(1) *Lines 25-26, change "living old trees, excavated archeological wood, old architectural wood, and naturally buried logs" to "living trees, archeological wood and buried logs"*

(2) Thank you for your comment.

(3) According to your comment, we have simplified the description of wood types (line 24).

(1) *Lines 46-49, to my knowledge, any method cannot reserve low-frequency climatic signals of tree-ring width/density fully.*

(2) Thank you for your suggestion.

(3) We have added a sentence according to your suggestion (lines 48-49).

(1) *Line 65 delete "summer"*

(2) Thank you for your comment.

(3) We have deleted "summer" (line 65).

(1) *Lines 67-68, cite studies with long tree-ring d18O chronology in Asia, such as Liu et al., 2017*

(2) Thank you for your comment.

(3) We have cited the paper of Liu et al. (2017) here (line 69).

(1) *Lines 69-84 describe methods on cellulose extraction and removing age trends. It's better to move them in Section 2.*

(2) Thank you for your suggestion. But, these descriptions are necessary to show our strategy of this study here (introduction).

(3) We have added relating description at the beginning of Section 2.2 (lines 102-103).

(1) *Line 94, hundreds of rings*

(2) Thank you for your comment.

(3) We have modified the word according to your suggestion (line 97).

That is all. Thank you for your comments.

Responses to Referee #2's comments.

We will present our responses to the referee's comments one by one in the following order. (1) *comments from Referees*, (2) author's response, (3) author's changes in manuscript (lines in the revised manuscript).

(1) *SYNOPTIC COMMENT The authors use of d2H and d18O tree-ring series for reconstruction of central Japan hydroclimate. Combining composite data set constitutes a serious technical challenge. The authors selected stem segments mostly of Japanese cypress from living trees, excavated archeological wood, architectural wood and naturally buried logs.*
*They propose an iterative calculation method to merge 67 series from the various types of wood samples, including the buried archeological and construction wood pieces, and a tentatively quantitative method (factors A and B) to calculate past climate based on d2H and d18O values, using a suite of equations derived from Roden's et al (2000).*
(2) Thank you very much for taking a lot of times to review this manuscript and give us valuable comments.
(3) We have revised our manuscript considering your valuable comments as below.

(1) *However, they did not nitrate their samples prior to analysing the d2H values of tree-ring cellulose, so that the exchangeable H is included in their analyses. The simple determination of d2H values on cellulose can generate artefacts.*
(2) In fact, we did not nitrate our cellulose samples, but there were two reasons. First, it was impossible to nitrate more than 10,000 samples of very small tree-ring cellulose in this study, because nitration of cellulose is very time-consuming and it usually needs more than 10 times larger amount of cellulose compared to the direct isotopic analyses of cellulose. Second, it was desirable to measure d18O and d2H simultaneously for the same cellulose samples in order to integrate d18O and d2H data in this study, but nitration of cellulose makes it impossible to measure the d18O. Indeed, measurement of the OH-hydrogen together with C-H hydrogen in cellulose may reduce the analytical precision of cellulose d2H to a certain degree, but it does not change the temporal pattern of d2H variations, to be compared with those of d18O, because all OH-hydrogen of cellulose in a wood segment must be exchanged with the same water in a test tube during the cellulose extraction process in this study. Moreover, we think that the negative influence of lower precision in d2H measurement was minimized in this study by focusing only on low-frequency component in d2H data, which can smooth analytical uncertainties in the individual treering cellulose d2H measurements.

(3) We have explained carefully the reason why we didn't nitrate samples in Section 2 (lines 126-130).

(1) *Additionally, the number of trees studied for d2H results are significantly lower than for the d18O determination, and the expressed population signals obtained for the composite d2H series are too low (Fig. 3, b, d). The fact that the d2H and d18O series do not derive from the same populations of trees may generate artefacts.*

(2) This is obviously a misunderstanding of the referee. The samples for d2H measurement in this study were completely same as those for d18O measurement. The low EPS for d2H series is owing to the low R-bar for d2H series. The core idea in this study is to measure both d18O and d2H simultaneously for all tree-ring cellulose samples and integrate them.

(3) In order to avoid misunderstandings, we have emphasized that we measure d18O and d2H simultaneously for all tree-ring cellulose samples in Section 2 (lines 112-114).

(1) *Another point is that the authors did not evaluate the reliability of the isotopic signals for the buried pieces of wood, but alteration of cellulose can occur due to microbial activities during long periods of burial.*

(2) Thank you for your comments. Indeed, tree-ring isotope ratios in buried wood are sometimes influenced by microbial activities in soil. However, we selected only well-reserved conifer woods from buried samples in this study. Moreover, we had confirmed that our method of cellulose extraction makes it possible to recover past cellulose isotopic ratios precisely even in the case of highly degraded hardwood samples by comparison of cellulose isotope ratios between degraded and non-degraded parts within a wood segment.

(3) We have added relating information in Section 2-2 (lines 107-109).

(1) *Overall, the article is lengthy for what it brings, but generally clearly written. The discussion of the low-frequency trends (long-periodicity variations) is confusing. The authors interpret them unguardedly as age trends, without presenting supporting arguments, and then they bring up the option of these trends possibly relating to changes in growth rates (lines 149-150; 160-165). This potential interpretation implies that environmental conditions may have generated these trends, at least partly. Moreover, the use of ring width for specifically deducing the cause of inverse d2H and d18O trends is risky because in many cases, the isotopic and ring width series do not respond to the same environmental factors.*

(2) Thank you for your comments. In this study, we demonstrated that changes in "the rate

of post-photosynthesis isotope exchange with xylem water" (f-value in this study) underlie low-frequency trends of tree-ring cellulose d18O by comparison of long-term variations in cellulose d18O and d2H based on the theory of Roden et al (2000), and suggested that changes in the growth rate may cause the changes in "f-value" by comparison of long-term variations in d18O and d2H with those in tree-ring width. As you suggest, it is impossible to remove "growth rate-related signals" from tree-ring cellulose d18O using tree-ring width data, because controlling factors are completely different between tree-ring width and tree-ring cellulose d18O. Instead, we use the data of tree-ring width here only for discussing the reason why low-frequency isotopic signals sometimes become opposite between d18O and d2H, and we don't use the data of tree-ring width for any kinds of quantitative calculation, so that the tree-ring width does not influence the long-term climate reconstruction in this study at all.

(3) We have added an Appendix to describe about the role of tree-ring width in this study carefully to avoid any kind of misunderstandings (Appendix A, Figure A1).

(1) *Another important point is that some of the sampled populations of trees belong to forests exposed to human perturbations; such sites are not suitable for producing isotopic series to be used for climatic reconstruction.*

(2) So far, dendroclimatologists have been thinking that trees exposed to human perturbation are not suitable for climate reconstruction. However, there are only few regions in the world where millennial length of non-human perturbed wood samples can be collected, such as Arctic region, mountainous regions in America and Eurasia etc. In most of other areas like Mediterranean and East Asian regions including Japan, where climatic influences to human history should be investigated using high-resolution paleoclimatological records over last several millennia, purely natural forests had already disappeared more than a few millennia ago owing to the intense logging activities by human beings. For the tree ring study in those areas and periods, most of wood samples are buried wooden artefacts excavated from archaeological remains, where human activities inevitably influenced the wood formation. Therefore, we believe that it is very important to establish a new sophisticated method for reconstruction of past climate variations using woods exposed to human perturbations in order to develop paleoclimatology in the world. The method proposed in this study is exactly to contribute to that purpose. Hence, we cannot accept this referee's comment at all, because this comment does not fit the real purpose of this study and prevents us from contributing to development of paleoclimatology in Japan and many other regions in the world.

(3) We have emphasized that in this study, we develop a dendroclimatological method to

utilize woods exposed to human perturbations effectively at many places (lines 90-91; 407-409; 571-574 etc).

(1) *Concluding that (1) d2H analyses would be more reliable if performed on nitrated cellulose, and (2) memory effects occur when performing online pyrolysis, are not new findings and do not bring constructive information in this field of research. Furthermore, as mentioned above, it underlines the fact that 50% of the data used for evaluating paleoclimate is faulty, and weakens the basis for the final reconstruction. Overall, given the purpose of the article and the unfortunate non-rigorous sample selection and treatment, CP should not accept this article.*

(2) We agree that the precision of d2H measurement in this study is not the best one in the viewpoint of analytical chemistry and it is obvious that the report of memory effect is not the purpose of this study. However, in fact, after receiving this comment, we have checked past many data carefully again and found that memory effect was not larger than reproducibility ($1\sigma$) of d2H measurement, so that it was not necessary for us to emphasize the existence of memory effect in this paper. We are very sorry to have confused you. Anyway, it was obvious that we cannot finish this study within a practical research period such as several years if we apply "nitration of cellulose to remove OH-hydrogen" and "triplicate measurements for individual samples to prevent memory effect", recommended by the referee, for the analyses of more than 15,000 tree-ring samples in this study, because those procedures request us to spend more than 10 times of analytical periods and sample amounts. In this study, we utilize the d2H data only for their low-frequency components smoothing of individual d2H data, so that we think that influence of possible lower precision in the individual d2H measurement must be minimized in the final result of this study. Of course, we agree that it is very important to develop "more sophisticated, practical and precise analytical method" of cellulose d2H. In order to promote development of the new analytical technology in the isotopic dendrochronology, we believe that it is very effective to publish this study utilizing d2H together with d18O for reconstructing of low-frequency climate variations explicitly, because recently the d2H is not often measured compared to the d18O in the field of isotopic dendrochronology.

(3) We have deleted the sentences on influence of memory effects and explained why we utilized our analytical procedures in Section 2.2 (lines 125-130).

SPECIFIC POINTS

(1) *Line 30 – replace However by In addition.*

(2) Thank you for your suggestion.

(3) We have replaced the word according to your suggestion (line 29).

(1) *Line 75 – Please provide a minimum of details about the direct cellulose extraction of 1-mm thick wood samples so the reader does not have to read two other articles to find out. Do you mean that all stem segments were dissected into suite 1 mm-thick samples regardless of ring width and age?*

(2) Yes. We sliced all stem segments into 1mm-thick wood plates regardless of ring width and age in order to keep the condition of cellulose extraction constant.

(3) We have added sentences and information on analytical details relating to the method in Kagawa et al (2015) (lines 104-107 and Fig. S2).

(1) *Line 76 – Please explain what are 'level offsets'.*

(2) Here, we use the "level offsets" to explain phenomenon that averaged tree-ring isotope ratios are significantly different between different individual trees during a same period.

(3) We have added some words to explain it in the manuscript (lines 77-78).

(1) *Line 80 – Simultaneous (?) measurements of d2H and d18O values? How possible with good precision? Using more than one standards is required for a good calibration (two end members with distant isotopic values defining a range broader than the measured isotopic ranges, and a third standard as an intermediate checkpoint), but the described analytical procedure does not mention this required approach.*

(2) We understand that the methods you recommend are effective to determine absolute d2H and d18O values of individual tree-ring cellulose precisely. However, according to the following 4 reasons, we decided to measure d2H and d18O "simultaneously (by peak jumping method)" using "only one standard material". 1) We must finish all d2H and d18O measurements of more than 15,000 tree-ring cellulose samples within practical research period of several years. 2) In order to integrate d2H and d18O data using the method in this study, it is highly desirable to obtain those data from completely same samples for individual tree rings, so that the simultaneous measurement of d2H and d18O for the same cellulose is the best way. 3) Large memory effects inevitably occur when multiple cellulose standards with distant d2H values are inserted in the sample measurements. 4) In this study, we discuss only the relative variations in d2H and d18O without considering their absolute values, so that the absolute precisions of measurements do not influence the main research result.

(3) We have added those explanations in the Section 2 (lines 112-114; 117-120; 127-130).

(1) *Lines 86-87 – Please modify text: : : for reconstructing climate over the past 2,600 yr: : :*

(2) Thank you for your suggestion.

(3) We have modified our manuscript according to your comment (line 89).

(1) *Lines 91-96 what are the average, minimum and maximum ring widths of the studied samples; this information will help follow the wood slicing procedure of next section.*

(2) Thank you for your suggestion.

(3) We have added the description about them in the caption of Fig. S2 together with a picture where we cut a narrow "cellulose" ring under a microscope. All information must be helpful to understand our procedure.

(1) *Lines 96-97 – Please briefly explain how the new tree-ring d18O time series were used for dating rings. Usage of a statistically strong constructed and multiply verified d18O suite as dating method? How widely is this applicable? For which geographical area was the dating series constructed? What is the operating time resolution on which the comparison is used?*

(2) Thank you for your comments. As you can see in Fig.3a, R-bar of tree-ring cellulose d18O is around 0.6-0.7 within the studied region in Fig.1, meaning that the cellulose d18O can be used not only to date all tree rings in this study precisely by the standard cross-dating method, but also to date any tree rings of any tree species collected in this region by comparison with the combined d18O chronology in Fig. 8 or 12 (Fig. 7 or 11 in the revised manuscript). In fact, those data are now being applied to date many tree rings not only in central Japan but also in western Japan and southern Korea because there are significant correlation (0.2-0.4) even between those distant areas.

(3) We have explained briefly about the dendrochronological dating in this study and the present situation of oxygen isotopic dendrochronology in Japan (Fig. S1 and its caption).

(1) *Lines 102-107 – Scientists have recognized for a long time that the production of tree ring d2H series to be coherent requires nitrating cellulose, so that only the C-bond hydrogen is analysed (Epstein et al., 1976; a reference they use and list). Otherwise, exchangeable H may blur true environmental effects. It seems here that the authors have chosen to save time by analysing simultaneously d2H and d18O in non-nitrated cellulose. No surprise that they conclude they need to improve their analytical procedure (lines 485-487). However, this information has been available to scientists since 1976, with a methods proposed for improving the throughput and reducing the amount of material required back in 2006 and*

*2009 (Filot et al., 2006; Sauer, 2009; rapid comm. mass spectrom.).*

(2) Thank you for your comments. As we mentioned above, there were two reasons why we did not nitrate cellulose ("saving of time and sample amount" and "necessity of simultaneous measurement of d2H and d18O"). Besides, we also understand that, under the two conditions, the method of Filot et al (2006) may be a good solution to obtain absolute d2H values of C-H hydrogen. However, we decided not to use the method of Filot et al (2006) according to the following two reasons. 1) We had already modified our auto-sampler system of TCEA to make it possible to measure about 200 cellulose samples per a day continuously, but the system of Filot et al (2006) could not be set to our system. 2) In this study, we do not need absolute d2H data, but only focus on relative variation in d2H for each individual stem segment, which can be obtained easily by our direct cellulose extraction method from a wood lath.

(3) We have explained those reasons briefly in Section 2.

(1) *Line 105 – Strict rigor would require indicating the true significant numbers for precisions (reproducibility), i.e., 0.1 or 0.2‰ and 1 or 2‰ for d18O and d2H values, respectively. In the light of the moderate correspondence between nitrated cellulose and cellulose, and of the analytical protocol (only one standard, memory effects not dealt with, peak jumping), it seems hardly conceivable that the d2H precision and accuracy would be of 1‰ it is likely no better than 3‰. Even with limited effects from OH-exchangeable fraction, the analytical precisions are rarely better than 2‰ (Filot et al., 2006; Sauer, 2009).*

(2) We agree to your comments in the sense of absolute accuracy. But, in this study, we only discuss about relative variations in d18O and d2H, and the measured cellulose d2H values in this study is affected by unknown OH-hydrogen d2H values which are constantly exchanged with experimental water during the cellulose extraction process, so that we do not discuss about the absolute accuracy but describe only the reproducibility of measurements here.

(3) We have explained this strategy briefly in Section 2.

(1) *Equations 1 and 2 – The ‰ sign should be on the left of the equations, near the delta notation. Otherwise, ‰ x 1000 implies no change in the reported values.*

(2) Thank you for your comments.

(3) We have modified those equations according to your comment (lines 121-122).

(1) *Lines 112-113 - Why all the cellulose samples could not be nitrated? Not enough material extracted from wood? The authors decided to follow an alternative approach, not clearly*

*defined (temperature, time of equilibration), but apparently different than the Filot approach, so that their cellulose and nitrated cellulose only show correlations (r) between 0.74 and 0.77, which is significantly lower than the correspondence obtained using the rigorous protocol of Filot et al (0.94). This compromise is not ideal when producing d2H series destined to climatic reconstruction.*

(2) As we mentioned above, we did not nitrate cellulose samples in order to same time and sample amounts and measure their d2H and d18O simultaneously. If we nitrated more than 15,000 samples in this study, we could not finish their measurements within a practical research period. In fact, the correlations between nitrocellulose d2H and cellulose d2H in this study were lower than that by Filot et al (2006). However, those correlation coefficients (0.77, 0.77) mean that we can infer d2H variations to a certain degree even using our method, and the fact that there are significant correlations between d18O and d2H in Fig.4c, 5c (Fig. S3c, S4c and S5c) and Fig.9 (Fig. 8 in revised one) suggests that we can obtain climatologically significant signals even using our method.

(3) As we mentioned above, we have explained carefully about our strategy of d2H measurement in Section 2.

(1) *Lines 118-119 – The authors should revisit this statement and write with more nuance, because they sacrifice on the reliability of the d2H series by analysing cellulose instead of nitrated cellulose, or by apparently using an alternative protocol that unfortunately does not perform as well as previous equilibration protocols documented in the literature (Filot et al., 2006; Sauer, 2009).*

(2) Thank you for your comments.

(3) As you point out, we have mentioned here more carefully about the possibility that the lower accuracy of d2H measurement in this study influence the data analyses (lines 135-139).

(1) *Lines 119-121 – The memory effects are well known when dealing with online pyrolysis systems, and there are several ways to avoid analytical artefacts due to them. Possibilities include placing a blank (empty capsule) between each samples in the carousels, or analyzing samples in triplicates, or a combination of the two approaches, etc. The appropriate analytical protocol with the instrument should be decided upfront, prior to producing the results. Unfortunately, again, the authors underline the issue after conducting all analyses, but truly this issue could have been easily dealt with prior to producing the isotopic series.*

(2) We understand several methods to reduce the memory effect as you recommend.

However, if we apply triplicate measurements for individual tree-ring cellulose and insert empty capsule between different samples, the total period necessary to finish more than 15,000 tree-ring measurements becomes 3-4 times longer. It seemed obviously unrealistic. In this study, the main research target is d18O, and the d2H is just used to remove long-term physiological effects in the d18O variations, so that we set the analytical conditions of tree-ring cellulose isotope ratios to maximize the efficiency of d18O measurements. By the way, after receiving of this comment, we have checked our past data again carefully and found that memory effects in d2H were not larger than the reproducibility (1σ) of d2H measurement, so that we realized that it was not necessary to describe the memory effect in detail. We are sorry to confuse you.

(3) We have deleted most of the explanation of memory effect in the draft.

(1) *Lines 125-129 – This text and Figure 3 do not inform the reader about the distribution of the wood types. Which isotopic series derive from buried pieces of wood? Departures from real values are reported to occur for altered cellulose/wood (Yapp, 2001; Mancini et al., 2003; Savard et al., 2012). Mancini, S.A., Ulrich, A.C., Lacrampe-Couloume, G., Sleep, B., Edwards, E.A., Lollar, B.S., 2003. Carbon and hydrogen isotopic fractionation during anaerobic biodegradation of benzene. Applied and Environmental Microbiology 69, 191–198. Yapp, C., 2001. Rusty relics of earth history: iron(III) oxides, isotopes, and surficial environments. Annual Review of Earth and Planetary Sciences 29, 165–199. Savard, M. M., Bégin, C., Marion, J., Arseneault, D., and Bégin, Y., 2012. Evaluating the integrity of C and O isotopes in sub-fossil wood from boreal lakes, Palaeogeogr. Palaeoclim. Palaeoecol., 348–349, 21– 31.*

(2) Thank you very much for introducing many papers on the effect of microbial wood degradation to their isotope ratios. You can see the "wood types" of all analyzed wood segments in Table 1. As for the description in Fig.3, all data before 5[th] century AD are owing to buried woods so that we don't think that it is meaningful to use different colors depending on wood types in Fig. 3. All woods analyzed in this study are conifer woods those are not seriously affected by microbial degradation, and moreover, we have confirm that our method of cellulose extraction can provide us of original cellulose isotope ratio even in the case of seriously degraded hardwood samples by comparison of isotope ratios of extracted cellulose between degraded part and undegraded part in a wood segment, so that we think that our data are not influenced by microbial alteration of wood isotope ratios.

(3) We have mentioned about influence of microbial degradation by referring related papers in Section 2 (line 107-109).

(1) *Lines 129 – What are the indications that these are age trends? Are trends visible on all Individual tree segments prior to combining them? Or are they visible after combining them? In the later case, the authors should consider discussing the possibility of an artefact to the treatment of the data.*

(2) In this paper, we use the words "age trend" just to indicate apparent long-term trend in the tree-ring isotope ratios, showing that the d18O gradually decreases with ages in most trees (Fig.3a). In fact, the long-term decrease in d18O is corresponding to that in the tree growth rate (tree-ring width), and the d18O sometimes increases suddenly even in old trees if the growth rate increases suddenly due to some environmental disturbances as shown in Fig. 4 and 5 (Fig. 5 and S3, S4, S5 in revised manuscript). And of course, if a long-term climate signal such as gradual drying overlaps the long-term d18O trend, we cannot see the apparent d18O decrease in that case. Therefore, both of your two questions are not right. All individual tree segments do not necessarily have the age trend, but most of them have the apparent age trends, and after combing all data, there remain the long-term age trends. Of course, it is not an artefact.

(3) In order to illustrate meaning of "age trends" clearly, we added Appendix A to describe a model to explain the mechanism of "age trends".

(1) *Lines 136-139 – The authors clearly state here why their d2H series are not reliable or suitable for climatic reconstruction. Ideally, they should not be used in the following parts of the article (or in the article).*

(2) We think that there are two possible reasons why R-bar in d2H is lower than that in d18O. One is the possible lower accuracy of d2H measurements in this study. But, as we mentioned above, memory effects were not actually serious. The other is the more complex mechanism in the post-photosynthesis isotope exchanges with xylem water for hydrogen than for oxygen as Roden et al (2000) described. Given that most of samples in this study are randomly affected by human perturbations, the complex nature of post-photosynthesis d2H alterations may lower the R-bar among tree segments in this study. Although it is not easy to utilize such data fully, we think that it is worth utilizing them to calibrate long-term age trends in cellulose $\delta^{18}O$ by restricting the usage only for the calibration.

(3) We have described the second reason as the main reason in the revised manuscript (line 154-155).

(1) *Section 3.2 – The entire discussion about the supposed 'age trends' is misleading. What are the arguments supporting this interpretation? If growth rates correlate with d2H and d18O, inversely and directly, what are the most logical environmental reasons for that?*

*Why do d2H and d18O trends inversely correlate (assuming that the d2H trends reflect something real)? Any possible mechanisms in teleconnection that could explain coeval long-term changes in the three proxies (growth, d2H and d18O)?*

(2) The "age trends" in this paper is corresponding to the apparent long-term gradual decreases in d18O, which have been frequently discussed in Esper et al. (2010) etc. However, we think that "apparent age trend" is not "real age trend" but caused by a physiological mechanism that the rate of post-photosynthesis isotope exchange with xylem water (f-value in this paper) gradually increase with age in some conifer trees. We also suppose that the increase in f-value is caused by the decrease in growth rate with age. This is a new hypothesis, but we think that it is sufficiently reasonable due to the following two reasons. 1) There are often opposite long-term trends between d18O and d2H variations, as shown in Fig. 3, 4 and 5 (S3, S4 and S5 in revised ones), which can be explained systematically only by the change in f-value as shown in Roden et al (2000). 2) In most cases when d18O and d2H suddenly changes to opposite directions in old trees, the tree-ring width also drastically changes, suggesting that the drastic changes in tree physiological conditions occur and cause the change in f-value through some biochemical mechanisms such as change in the utilization rate of stored carbohydrate for cellulose synthesis. Although this phenomenon has not been discussed in the tree-ring studies, we have already published a paper on tree physiology to demonstrate the existence of this mechanism by comparison of intra-ring variations in cellulose d18O and d2H (Nabeshima et al., 2018).

(3) In order to describe the meaning and mechanism of "apparent long-term age trend" in tree-ring cellulose d18O in Japanese conifer in detail, we have added Appendix A (Leaf volume model for long-term age trends in tree-ring cellulose d18O and d2H of Japanese conifers).

(1) *Lines 214-215 – It seems that the sites selected for this research are not suited for climatic reconstruction.*

(2) We cannot accept this comment as mentioned above. Because this is the most important characteristic of this paper, we will answer to this comment again. So far, dendroclimatologists have been thinking that trees exposed to human perturbation are not suitable for climate reconstruction. However, there are only few regions in the world where millennial length of non-human perturbed wood samples can be collected, such as Arctic region, mountainous regions in America and Eurasia etc. In most of other areas like Mediterranean and East Asian regions including Japan, where climatic influences to human history should be investigated using high-resolution paleoclimatological records

over last several millennia, purely natural forests had already disappeared more than a few millennia ago owing to the intense logging activities by human beings. For the tree ring study in those areas and periods, most of wood samples are buried wooden artefacts excavated from archaeological remains, where human activities inevitably influenced the wood formation. Therefore, we believe that it is very important to establish a new sophisticated method for reconstruction of past climate variations using woods exposed to human perturbations in order to develop paleoclimatology in the world. The method proposed in this study is exactly to contribute to that purpose. Hence, we cannot accept this referee's comment at all, because this comment does not fit the real purpose of this study and prevents us from contributing to the development of paleoclimatology in Japan and many other regions in the world.

(3) We have emphasized that in this study, we develop a dendroclimatological method to utilize woods exposed to human perturbations effectively at many places (lines 90-91; 407-409; 571-574 etc).

(1) *Lines 251-252 & Eq 18 – The physiological effects does not always generate a negative relationship between d2H and d18O series. Is it not right?*

(2) Thank you for your comments. Of course, as you point out, all of physiological mechanisms do not necessarily change the d2H and d18O to opposite directions. The physiological effect defined in this study is related only to the changes in the rate of post-photosynthesis isotope exchange with xylem water before cellulose synthesis (f-value). Other biological mechanisms related to "age effect" such as "root deepening" are not considered in Eq.18, so that they are unfortunately added to the climatological component in Eq. 17, if such mechanism actually exists. Here, we propose the method to remove the physiological component from d18O variations based on the assumption that physiological effects other than "change in f-vale" are negligible.

(3) In order to avoid misunderstandings, we have clearly explained that we focus only on the changes in the post-photosynthesis isotope exchange with xylem water (f-value) as the cause of physiological influence to the tree-ring d18O and d2H variations (lines 256-259).

(1) *Lines 258 & 267 – Using constant A and B values implies multiple big assumptions.*

(2) We deeply agree to your comments. In fact, adequate A and B values may be different for different individual trees, and it may become possible to propose different A and B values for individual trees by some sophisticated methods in near future. However, in this paper, we decided to fix A and B values constant according to the flowing reasons in addition to the convenience in calculation procedure already described in the text. 1) The

method integrating d18O and d2H to extract climatological components proposed in this paper is a totally new one and very complicated even in the present condition. If we add the procedure to set different A and B values for individual trees in this paper, the paper become lengthier and cannot be understood easily. 2) We think that it is meaningful to investigate the resultant long-term variation in climatological component of d18O extracted using constant A and B values as their simplest cases at first in order to develop this method further. In fact, as we demonstrate in Section 3.11, the resultant long-term variation in climatological component of d18O are well corresponding to those in other lower-resolutions of paleoclimate proxies, suggesting that the assumption of constant A and B values has proven realistic to a certain degree. We anticipate that it promotes the participation of many other researchers to the study of the d2H and d18O relationships.

(3) We have explained about the reason why we set A and B constant here (lines 274-278).

(1) *Line 296-298 – Another big assumption that this simple combination cancels out the inter-tree average offsets.*

(2) Thank you for your comments. The biggest assumption in Line 296-298 is related to the determination of B value in Fig. 10. But the inter-tree average offset itself can be cancelled out explicitly by the method described in Fig.6.

(3) We have modified the sentence to make the meaning of assumptions clearer (lines 306-308).

(1) *Section 3.7 – How can the authors attest that this approach does not generate artefacts at the point of junction between series (e.g., Gagen et al., 2012). Gagen, M., McCarroll, D., Jalkanen, R., Loader, N. J., Robertson, I., and Young, G. H. F., 2012. A rapid method for the production of robust millennial length stable isotope tree ring series for climate reconstruction, Global Planet. Change, 82–83, 96-103.*

(2) As you can see in Fig.6, the iterative calculation in the averaging and offsetting method finally makes a combined time-series where its averaged d18O value during the period corresponding to an individual tree segment becomes equal to that of the offset d18O variation of the individual tree segment. If you start from the same original dataset on d18O variations of tree segments, the pattern of relative variation in the final combined d18O time-series is mathematically unique although its absolute value has no meaning. That is, there is no room where some artefact influences resultant time-series.

(3) We have added this explanation briefly in Section 3.7 (line 320-323).

(1) *Section 3.8 – It seems that there are several short cuts slid in the procedure for attempting*

*to correct for limitations introduced by the analytical approach (lines 367-371). Since there is no true comparison with a fully rigorous approach, the assessment of the procedure is impossible. The comparisons made with reconstructions from other proxies show significant departures and do not allow assessing the proposed procedure (section 3.11).*

(2) As illustrated in Fig.4 and 5 (S3, S4, S5 in revised ones), the physiological effect, defined in this study, influences to the low-frequency component of d18O only, so that it is not necessary to integrate d2H and d18O in the high-frequency component for removal of the physiological signals in d18O. If the lower R-bar in d2H is caused by its low analytical precision due to non-nitration, we can assume that it influences the d2H data randomly, so that smoothing of d2H data to make low-frequency d2H variation can minimize the negative effect of the lower precision of d2H measurement. That is the reason why we selected the calculation procedure in Section 3.8 (Fig. 6). Given that all low-frequency paleoclimate reconstructions referred in Section 3.11 were obtained from different spatial scales using completely different proxies, it is reasonable that there are some discrepancies from the result obtained in this study, but the overall similarities in the low-frequency components suggest that there are certain significances in the dataset and calculating procedures in this study.

(3) In order to make clear the meaning of calculation procedure, we have modified the sentences carefully (Section 3.8). In the previous manuscript, the comparison of the climatological components in tree-ring cellulose $\delta^{18}O$ (reintegrated $\Delta\delta^{18}O_{cel(climate)}$) with existing high- and low-frequency climate and paleoclimate records in Section 3.11 was only one way to estimate the reliability of our strategy to extract climatological components in tree-ring cellulose $\delta^{18}O$. In this revised manuscript, we add a new figure (Fig. A3) in Appendix A to demonstrate the effectiveness of calculating $\Delta\delta^{18}O_{cel(climate)}$ and reintegrated $\Delta\delta^{18}O_{cel(climate)}$ by comparison of three very old trees covering from 12$^{th}$ to 20$^{th}$ centuries.

(1) *Line 359 – Note clear... as: : : Please rewrite.*

(2) Thank you for your suggestion.

(3) We have modified the sentence carefully by adding words to make it clear (lines 373-374).

(1) *Lines 394-396 – The idea is with paying attention, but unfortunately, the basic sampling and analytical procedures selected for this research are not rigorous enough to allow evaluating the approach in this article.*

(2) As we mentioned above, one of the most important purpose of this study is to propose a method to reconstruct multi-millennial climate variations using wood segments highly affected by human perturbation. In order to analyze both d18O and d2H of more than

15,000 tree-ring cellulose samples within practical research period, we needed some compromises on the analytical procedures, but the sampling strategy of woods exposed to human perturbation was not an unnecessary fault but the essential part of this study. However, as you mention, there remain many points which should be improved for further development of this method.

(3) According to your comment, we have added some sentences (lines 499-508).

(1) *Table 1 and Figure 2 – It seems that the term 'sample' here refers to stem segments.*

(2) Thank you for your suggestion.

(3) We have replaced the word or deleted the term 'sample' according to your suggestion (Table S1 and Figure 2).

(1) *Figure 2 – 70% line? Not clear what it is and what it means?*

(2) Thank you for your suggestion. Yes, it is not clear.

(3) We have deleted the word "70% line" in the figure and added the explanation of the line in the figure legend (Figure 2).

(1) *The number of figures is high; perhaps some of them would find a better place in a supplement of information, for examples figures 9, 13, 14.*

(2) In fact, there are many figures, but we think that all of Figs 9, 13 and 14 (Fig. 8, 12, 13 in revised ones) play important roles in this paper. So, if possible, we want to leave these figures at the position near the corresponding text.

(3) We have moved some figures (Figs S1, S2, S3, S4, S5) and a table (Table S1) to supplement in addition to some figures in Appendix (Figs A1, A2, A3).

That is all. Thank you very much for your valuable comments.

---

## Author Response (AR2)

29 Sep 2020

Dear Editor and members of the editorial support team,

I am sending herewith two PDF files of the corrected manuscript entitled as "A 2600-year summer climate reconstruction in central Japan by integrating tree-ring stable oxygen and hydrogen isotopes" and its supplementary materials.

Thank you very much for your positive decision, as shown below, toward the final publication of our article.

Editor Decision: Publish subject to technical corrections (21 Sep 2020)
by Hans Linderholm Comments to the Author:
Dear Dr Nakatsuka and co-authors
The reviewers reports are now in, and as you can see, they are both positive and suggest that your manuscript should be published. I agree, so well done. I will be happy to accept your work subsequent to you having gone through it again focusing on the English. Also. you mention figure S1 etc, but have included figures A1-3, so please be consistent. I'm looking forward to seeing the final version shortly.
Best regards
Hans

In fact, I could not find any reports other than this short message in the website. Although I have sent an e-mail message to the editorial support team to ask whether there are other files of reports or not, I have not received any replies until now.

Therefore, I tried to correct our manuscript according to this short message only. I think that there are two suggestions for the corrections.

1) Coexistence of "Supplement" and "Appendix" are contradictory.
I am very sorry to have confused you. In this corrected manuscript, contents in Appendix have been integrated to Supplement.

2) "English" should be corrected.
Once, I heard that the editorial support team of "Climate of the Past" will give us concrete advices on English corrections AFTER the final acceptance. So, we have not corrected

(improved) our English in the files I sent today, where sentences those have not passed the English correction are shown in red colored characters.

If our English should be corrected BEFORE the final acceptance by ourselves, we must request the extension of the deadline (29 Sep 2020) at least one or two week, because we need to ask some private company of English editing and it takes about one or two weeks for us to get the corrected version from the private company and fix the corrected parts.

Thank you very much for your understanding. It is not easy for Japanese people to finalize English sentences by themselves due to the large language gap between English and Japanese.

Best regards,
Takeshi

Takeshi Nakatsuka, PhD. Professor
Graduate School of Environmental Studies, Nagoya University, Japan